# A Theoretically-Principled Sparse, Connected, and Rigid Graph Representation of Molecules

**Shih-Hsin Wang**[1*]**, Yuhao Huang**[1*]**,Justin Baker**[2]**, Yuan-En Sun**[3]**, Qi Tang**[4] **& Bao Wang**[1]

[1]Department of Mathematics and Scientific Computing and Imaging (SCI) Institute
University of Utah, Salt Lake City, UT 84102, USA
[2]Department of Mathematics, UCLA, Los Angeles, CA 90095, USA
[3]Department of Biochemistry, University of Utah, Salt Lake City, UT 84102, USA
[4]School of Computational Science and Engineering, Georgia Tech, Atlanta, GA 30332, USA

## Abstract

Graph neural networks (GNNs) – learn graph representations by exploiting the graph's sparsity, connectivity, and symmetries – have become indispensable for learning geometric data like molecules. However, the most used graphs (e.g., radial cutoff graphs) in molecular modeling lack theoretical guarantees for achieving connectivity and sparsity simultaneously, which are essential for the performance and scalability of GNNs. Furthermore, existing widely used graph construction methods for molecules lack rigidity, limiting GNNs' ability to exploit graph nodes' spatial arrangement. In this paper, we introduce a new hyperparameter-free graph construction of molecules and beyond with sparsity, connectivity, and rigidity guarantees. Remarkably, our method consistently generates connected and sparse graphs with the edge-to-node ratio being bounded above by 3. Our graphs' rigidity guarantees that edge distances and dihedral angles are sufficient to uniquely determine the general spatial arrangements of atoms. We substantiate the effectiveness and efficiency of our proposed graphs in various molecular modeling benchmarks. Code is available at https://github.com/shihhsinwang0214/SCHull.

## 1 Introduction

Graph neural networks (GNNs) have emerged as indispensable tools for learning graphs, e.g., social networks (Perozzi et al., 2014) and molecules (Duvenaud et al., 2015; Gilmer et al., 2017). GNNs operate by exchanging messages between neighboring nodes for learning features at both node and graph levels; see, e.g. (Gilmer et al., 2017; Satorras et al., 2021). Constructing sparse and connected graph representations of the input data plays a pivotal role in the success of GNNs, especially in molecular modeling. Graph sparsity enables computational efficiency, and connectivity ensures that information can flow seamlessly throughout the network – it has been shown that the disconnectivity of a graph can degrade the model's performance; see, e.g., (Sverdlov & Dym, 2024).

A common method for creating molecular graphs involves setting a radial cutoff distance; atoms are connected by edges if their Euclidean distance falls below the cutoff threshold (Schütt et al., 2017; 2018; Unke & Meuwly, 2019). To limit the number of connections per atom, ad-hoc post-pruning can be applied, removing edges between distant atoms if a node has too many connections. However, determining an optimal cutoff for molecular graphs presents a significant challenge. A large cutoff might guarantee that all atoms are connected, but it can result in overly dense graphs. Conversely, while post-pruning can reduce the density of graphs, it might introduce the risk of disconnectivity. Furthermore, for large molecules like proteins with thousands of atoms, a fixed cutoff is often insufficient to ensure connectivity. Examples include long-range atomic interactions in protein allosteric regulation (Süel et al., 2003; Zhu et al., 2022) and inter-chain interactions in protein oligomerization (Chen et al., 2021; Chen & Bell, 2021). To illustrate the limitations of the radial cutoff method, we present Fig. 1(a), which depicts a protein molecule with colored residues. These colored residues represent a single instance from the Fold biomolecule dataset (Murzin et al., 1995; Hou et al., 2018). As shown in Fig. 1(b), applying a cutoff of 10Å on the data consisting of $C_\alpha$ atoms of amino acids results in a disconnected but quite dense graph. To further solidify

---

*Shih-Hsin Wang and Yuhao Huang are co-first authors. Correspond to wangbaonj@gmail.com

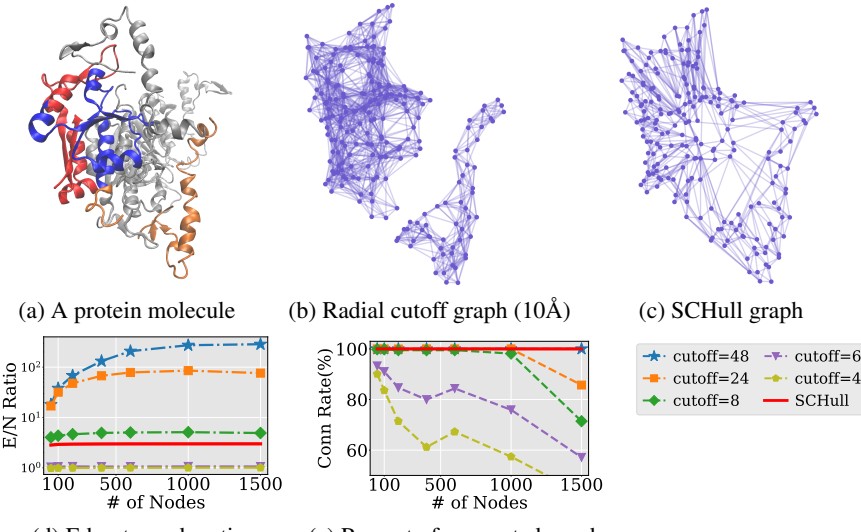

(a) A protein molecule     (b) Radial cutoff graph (10Å)     (c) SCHull graph

(d) Edge-to-node ratio     (e) Percent of connected graphs

Figure 1: Illustration of challenges of radial cutoff graphs in balancing sparsity and connectivity for the Fold biomolecule dataset. (a) Colored residues in a large protein, representing a single instance from the Fold dataset. (b) A radial cutoff graph with a cutoff of 10Å for the colored residues, demonstrating disconnectivity and density; in contrast, (c) showcases our proposed SCHull graphs, ensuring sparsity and connectivity. Panels (d) and (e) plot the number of nodes against the edge-to-node ratio and the percentage of connected graphs, respectively, for radial cutoff graphs using different thresholds and our SCHull graph construction. The results highlight that radial cutoff has to trade sparsity for connectivity, while our SCHull graphs maintain both properties.

the challenges in balancing sparsity and connectivity using radial cutoff graphs, we investigate the sparsity and connectivity tradeoff using various cutoff thresholds to construct graphs for proteins in the Fold dataset. Next, for these radial cutoff graphs, we calculate the edge-to-node ratio and the proportion of connected graphs for different cutoff thresholds. As shown in Fig. 1(e), a cutoff of 48Å is required to achieve connectivity for large proteins with more than 1000 amino acids. However, as shown in Fig. 1(d), connectivity comes at the cost of dramatically increased density, e.g., a cutoff of 48Å results in graphs with over 100 times more edges than graphs constructed with a 6Å cutoff. We further discuss other popular graph representations of molecules like chemical graphs and $k$-nearest neighbor graphs in Section 2.

Besides sparsity and connectivity, it's essential to consider whether a geometric graph could contain sufficient information to determine its nodes' spatial arrangement (geometries). Pairwise distances between all nodes theoretically determine the geometric arrangement (Satorras et al., 2021) up to rigid motions, but computing pairwise distances between all nodes amounts to involving a dense graph in the operation. Alternatively, researchers have explored sparse edge distributions with specific geometric features to identify the spatial arrangement of graph nodes. Sverdlov & Dym (2024) connect identifying nodes' arrangement to graph rigidity and have demonstrated that graph rigidity can ensure a GNN's ability to distinguish general geometric configurations of nodes. Nevertheless, radial cutoff graphs often lack sufficient rigidity, leading to ambiguity in spatial arrangements (Joshi et al., 2023; Wang et al., 2024). Figure 2 illustrates this limitation using a synthetic example: A and B are two

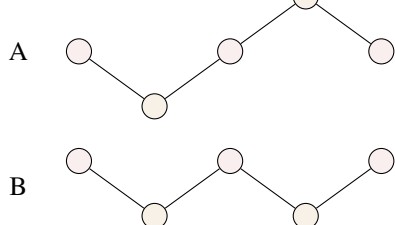

Figure 2: Illustration of two distinct, non-isometric arrangements of nodes in Euclidean space. When applying a radial cutoff to the arrangements in A and B, we obtain connected and sparse graphs. However, the corresponding edges in both graphs have identical lengths, making it challenging for graph neural networks to distinguish between them.

distinct, non-isometric arrangements of five points in Euclidean space; while applying a radial cutoff to these arrangements results in connected and sparse graphs. However, the edges in both graphs have the same lengths, making it challenging for GNNs to distinguish between them.

## 1.1 OUR CONTRIBUTIONS

In response to the limitation of existing graph constructions of molecules, we propose Spherical Convex Hull (SCHull) – a new hyperparameter-free graph construction method tailored for data with 3D coordinates, especially molecules. SCHull constructs graph representations of the underlying

points in two computationally efficient steps: (1) project points onto the unit sphere centered at the center of points, and (2) construct a convex hull for the projected points and add an edge to connect two original points if the convex hull has an edge connecting the corresponding projected points; see details in Section 3.1. SCHull graphs enjoy a few advantages with theoretical guarantees:

- SCHull graphs are connected and sparse. In particular, the edge-to-node ratio is bounded above by 3; see detailed theoretical results in Section 3.2.
- SCHull graph mirrors the spatial arrangement of the underlying point clouds, ensuring that encoding edge distances and dihedral angles are sufficient for GNNs to uniquely determine the geometric configurations of nodes up to isometries; see Section 3.3 for details.

Furthermore, the SCHull graph can be seamlessly coupled with existing GNN learning techniques to boost the performance of off-the-shelf models. We substantiate the effectiveness and efficiency of the proposed SCHull graph using a few molecular modeling benchmarks, including predicting atomic forces for small chemicals, protein fold classification, enzyme reaction classification, and protein-ligand binding affinity prediction; see details in Section 4. Our numerical results show that using SCHull can consistently improve the prediction accuracy of existing remarkable GNNs by a significant margin at the cost of a tiny computational overhead.

## 1.2 ORGANIZATION

We organize this paper as follows: In Section 2, we recap on necessary background materials and some related results. We present the construction process of our proposed SCHull graphs in Section 3.1, followed by their sparsity and connectivity analysis in Section 3.2. We further analyze the rigidity of SCHull graphs and their benefits in improving GNN's performance in Section 3.3. We substantiate the performance of SCHull in Section 4 using various benchmark tasks in molecular modeling. Technical proofs and additional experimental details are provided in the appendix.

## 2 BACKGROUND AND SOME RELATED WORKS

In this section, we provide a brief review of key concepts and related results. Specifically, we revisit concepts of geometric graphs as discussed by Joshi et al. (2023); Wang et al. (2024), review existing graph construction methods for molecules, and introduce convex hulls and polyhedral graphs along with a key result on their rigidity. An overview of graph rigidity is provided in Appendix B, along with a quick example. For a more in-depth exploration, we refer readers to (Asimow & Roth, 1978; Connelly, 2005). Furthermore, we will recap on message-passing GNNs and their separation power.

**Point clouds and geometric graphs.** A 3D *point cloud*, consisting of $m$ points $\boldsymbol{x}_1, \boldsymbol{x}_2, \ldots, \boldsymbol{x}_m$, can be viewed as a spatial arrangement of a set of $m$ nodes $\mathcal{V} = \{1, 2, \ldots, m\}$, where each index $i$ is associated with a node with coordinate $\boldsymbol{x}_i \in \mathbb{R}^3$. We can represent a point cloud as a set $\{\boldsymbol{x}_i\}_{i=1}^m$ or as a spatial arrangement of nodes $(\mathcal{V}, \boldsymbol{X} = [\boldsymbol{x}_1, \boldsymbol{x}_2, \ldots, \boldsymbol{x}_m])$. Optionally, each node may be associated with features $\boldsymbol{f}_i \in \mathbb{R}^{n_f}$ beyond coordinates. Extending from a point cloud to a *geometric graph* – denoted as $\mathcal{G} = (\mathcal{V}, \mathcal{E}, \boldsymbol{X})$ (resp. $(\mathcal{V}, \mathcal{E}, \boldsymbol{X}, \boldsymbol{F})$) – introduces a graph structure by connecting nodes using edges in the set $\mathcal{E}$ to include relationships between points in $(\mathcal{V}, \boldsymbol{X})$ (resp. $(\mathcal{V}, \boldsymbol{X}, \boldsymbol{F})$). Here, $(i, j) \in \mathcal{E}$ denotes an edge and $\boldsymbol{F} = [\boldsymbol{f}_1, \ldots, \boldsymbol{f}_m]$ represents node features.

**Existing graph representations of molecules.** As mentioned earlier, radial cutoff is one of the prevailing approaches to constructing graph representations of molecules. Besides radial cutoff, there are several other methods have also been used to construct graphs for molecules, including $k$-nearest neighbor graphs ($k$NN) (Jørgensen et al., 2018) and chemical graphs (Gilmer et al., 2017). Similar to the radial cutoff graphs, $k$-nearest neighbor graphs, and chemical graphs also struggle to achieve sparsity, connectivity, and rigidity at the same time; we contrast different graphs' properties in Table 1.

Table 1: Contrasting different graph construction methods in achieving sparsity, connectivity, and rigidity.

| Construction method | Sparsity | Connectivity | Rigidity |
|---|---|---|---|
| Radial cutoff (small threshold) | ✓ | ✗ | ✗ |
| Radial cutoff (large threshold) | ✗ | ✓ | ✓ |
| $k$-nearest neighbors | ✓ | ✗ | ✗ |
| Chemical graphs | ✓ | ✓ | ✗ |
| SCHull (ours) | ✓ | ✓ | ✓ |

**Rigid motions and geometric isomorphism.** A *rigid motion*, a.k.a. an *isometry*, is a transformation defined by an orthogonal matrix $\boldsymbol{Q} \in \mathrm{O}(3)$ and a translation vector $\boldsymbol{t} \in \mathbb{R}^3$. It acts on the point cloud

by transforming all its points according to $\boldsymbol{x}_i \mapsto \boldsymbol{Q}\boldsymbol{x}_i + \boldsymbol{t}$. The group of 3D rigid motions is called the Euclidean group and is denoted by E(3). Two geometric graphs $\mathcal{G}_1 = (\mathcal{V}_1, \mathcal{E}_1, \boldsymbol{X}_1, \boldsymbol{F}_1)$ and $\mathcal{G}_2 = (\mathcal{V}_2, \mathcal{E}_2, \boldsymbol{X}_2, \boldsymbol{F}_2)$ are said to be *geometrically isomorphic* if and only if there exists a bijection $b : \mathcal{V}_1 \to \mathcal{V}_2$ and a rigid motion $(\boldsymbol{Q}, \boldsymbol{t}) \in E(3)$ such that for any $(i, j) \in \mathcal{E}_1$, $(b(i), b(j)) \in \mathcal{E}_2$, and for any $i$, $\boldsymbol{f}_i = \boldsymbol{f}_{b(i)}$, $\boldsymbol{x}_i = \boldsymbol{Q}\boldsymbol{x}_{b(i)} + \boldsymbol{t}$. Notice that the condition requiring a rigid motion $(\boldsymbol{Q}, \boldsymbol{t}) \in E(3)$ such that $\boldsymbol{x}_i = \boldsymbol{Q}\boldsymbol{x}_{b(i)} + \boldsymbol{t}$ for all $i$ is equivalent to the condition $\|\boldsymbol{x}_i - \boldsymbol{x}_j\| = \|\boldsymbol{x}_{b(i)} - \boldsymbol{x}_{b(j)}\|$ for all $i, j$. A function $F$ defined on the geometric graphs is said to be *invariant* if $F(\mathcal{G}_1) = F(\mathcal{G}_2)$ for any two geometrically isomorphic geometric graphs $\mathcal{G}_1, \mathcal{G}_2$.

**Convex hull and polyhedral graph.** A *convex hull* of a set of points $\{\boldsymbol{x}_1, \ldots, \boldsymbol{x}_m\} \subset \mathbb{R}^3$ is defined as $\text{Conv}(\{\boldsymbol{x}_1, \ldots, \boldsymbol{x}_m\}) = \{\sum_i \alpha_i \boldsymbol{x}_i \mid \sum_i \alpha_i = 1, \alpha_1, \ldots, \alpha_m \in [0, 1]\}$, which is the smallest convex set containing these points. When the points are not coplanar, the convex hull is also known as a *convex polyhedron*, whose graph structure comprises its vertices, edges, and faces. Any graph derived from this structure is referred to as a *combinatorial polyhedron* or *polyhedral graph*. Steinitz's Theorem (Grünbaum, 2007) establishes a one-to-one correspondence between polyhedral graphs and 3-connected simple planar graphs[1]. This means every polyhedral graph can be represented as a 3-connected simple planar graph and vice versa.

A *geometric polyhedral graph* is a geometric graph $(\mathcal{V}, \mathcal{E}, \boldsymbol{X})$ whose underlying graph structure is a polyhedral graph. It can be considered as assigning node coordinates to a polyhedral graph $(\mathcal{V}, \mathcal{E})$. A well-known rigidity theorem for polyhedral graphs by Stoker (1968) (see also (Cho & Kim, 2023)) states that a strictly-convex geometric polyhedral graph is uniquely determined by its edge lengths $d_{ij}$ and dihedral angle $\tau_{ij}$ on edge $(i, j)$, where the dihedral angle on edge $(i, j)$ is defined as the angle between two faces that contain edge $(i, j)$. More precisely, we have the following theorem:

**Theorem 2.1** (Stoker (1968)). *Let $\mathcal{G} = (\mathcal{V}, \mathcal{E}, \boldsymbol{X})$ and $\mathcal{G}' = (\mathcal{V}', \mathcal{E}', \boldsymbol{X}')$ be two strictly-convex geometric polyhedral graphs. Suppose there exists a graph isomorphism $b : \mathcal{V} \to \mathcal{V}'$ satisfying $\tau_{ij} = \tau'_{b(i)b(j)}$ and $d_{ij} = d'_{b(i)b(j)}$ for any $(i, j) \in \mathcal{E}$, where $d_{ij}$ and $\tau_{ij}$ denote the length of edge $(i, j) \in \mathcal{E}$ and dihedral angle associated with edge $(i, j)$. Then $\mathcal{G}$ and $\mathcal{G}'$ must be geometrically isomorphic, i.e., there is a rigid motion $(\boldsymbol{Q}, \boldsymbol{t}) \in E(3)$ such that $\boldsymbol{x}_i = \boldsymbol{Q}\boldsymbol{x}'_{b(i)} + \boldsymbol{t}$ for any $i \in \mathcal{V}$.*

**Message-passing GNNs and their separation power.** For any geometric graph $\mathcal{G} = (\mathcal{V}, \mathcal{E}, \boldsymbol{X}, \boldsymbol{F})$, a message-passing GNN propagates features, with $\boldsymbol{f}_i^{(0)} = \boldsymbol{f}_i$, as follows:

$$\boldsymbol{f}_i^{(t+1)} = \text{UPD}\left(\boldsymbol{f}_i^{(t)}, \text{AGG}(\{\!\!\{\boldsymbol{f}_i^{(t)}, \boldsymbol{f}_j^{(t)}, e_{ij} \mid j \in \mathcal{N}_i\}\!\!\})\right), \tag{1}$$

where $e_{ij}$ are edge attributes ($e_{ij}$ can be $\boldsymbol{x}_i - \boldsymbol{x}_j$ or pre-determined invariant edge attributes like relative distance $\|\boldsymbol{x}_i - \boldsymbol{x}_j\|$), $\mathcal{N}_i$ denotes the one-hop neighbors of node $i$, i.e., the set of nodes in $\mathcal{V}$ that are reachable from $i$ through an edge in $\mathcal{E}$, and UPD and AGG are learnable update and aggregate functions operating on the graph $\mathcal{G}$, respectively. After $T$ message-passing steps, we apply a multiset Readout function to obtain the graph level global features

$$\boldsymbol{s}^{\text{global}} = \text{Readout}\left(\{\!\!\{\boldsymbol{f}_i^{(T)} \mid i \in \mathcal{V}\}\!\!\}\right). \tag{2}$$

A common approach to evaluating GNN's performance is to assess its ability to learn distinct global features from non-isomorphic geometric graphs. In particular, a standard assumption in this analysis is to consider maximally expressive GNNs, where the update, aggregation, and readout functions are all injective (Joshi et al., 2023; Wang et al., 2024; Sverdlov & Dym, 2024). We say a maximally expressive GNN $F$ with depth $T$ can distinguish two geometric graphs $\mathcal{G}_1$ and $\mathcal{G}_2$ if, after $T$ iterations, the learned global features for $\mathcal{G}_1$ and $\mathcal{G}_2$ are distinct, that is, $\boldsymbol{s}_1^{\text{global}} =: F(\mathcal{G}_1) \neq F(\mathcal{G}_2) := \boldsymbol{s}_2^{\text{global}}$.

## 3 MAIN RESULTS

In response to the limitation of existing graph representations – in achieving sparsity, connectivity, and rigidity – for geometric data with 3D coordinates, we propose SCHull – a new graph construction method for molecules and beyond. SCHull eliminates hyperparameter tuning and ensures connectivity while maintaining sparsity and rigidity. In this section, we will outline the two steps in constructing SCHull graphs, followed by proving how these steps achieve the desired properties and how the resulting graph benefits GNNs' performance.

---

[1]In this context, a simple graph is one without self-loops or multiple edges. A graph is 3-connected if removing fewer than three vertices doesn't disconnect it. A graph is planar if it can be drawn on a plane without edge crossings.

### 3.1 SCHull – A New Graph Construction

SCHull constructs graphs in two steps: (1) project points (nodes) onto the unit sphere centered at the center of the points, and (2) construct the convex hull for the projected points and connect two original points if the convex hull contains an edge connecting the two corresponding projected points.

**Step 1: Project points onto the unit sphere.** Let $(\mathcal{V}, \boldsymbol{X} = [\boldsymbol{x}_1, \boldsymbol{x}_2, \ldots, \boldsymbol{x}_m])$ be a point cloud, e.g., representing atoms in a protein molecule. Denote the center of $\boldsymbol{X}$ by $\overline{\boldsymbol{x}} := \frac{1}{m} \sum \boldsymbol{x}_i$. Consider the projection $p_{\overline{\boldsymbol{x}}} : \mathbb{R}^3 \to \mathbb{S}^2$ defined by $\boldsymbol{x} \mapsto \frac{\boldsymbol{x} - \overline{\boldsymbol{x}}}{\|\boldsymbol{x} - \overline{\boldsymbol{x}}\|}$, where $\mathbb{S}^2 := \{\boldsymbol{x} \in \mathbb{R}^3 \mid \|\boldsymbol{x}\| = 1\}$ is the unit sphere. That is, $p_{\overline{\boldsymbol{x}}}$ projects points onto the unit sphere centered at $\overline{\boldsymbol{x}}$. Applying this projection to all points, we obtain a new point cloud $(\mathcal{V}, p_{\overline{\boldsymbol{x}}}(\boldsymbol{X}))$ on $\mathbb{S}^2$, where $p_{\overline{\boldsymbol{x}}}(\boldsymbol{X}) = [p_{\overline{\boldsymbol{x}}}(\boldsymbol{x}_1), p_{\overline{\boldsymbol{x}}}(\boldsymbol{x}_2), \ldots, p_{\overline{\boldsymbol{x}}}(\boldsymbol{x}_m)]$.

In rare cases, there may be points such that $\boldsymbol{x}_i = \overline{\boldsymbol{x}}$ for some $i$, which prevents us from defining its projection. To ease our analysis, we assume that the projections of all points in a given point cloud $(\mathcal{V}, \boldsymbol{X})$ are well-defined and these projections on $\mathbb{S}^2$ are distinct. This implies that $|\mathcal{V}| = |\{p_{\overline{\boldsymbol{x}}}(\boldsymbol{x}_i) \mid i \in \mathcal{V}\}|$. More precisely, we assume the following *generic condition* holds:

$$\boldsymbol{x}_i \neq \overline{\boldsymbol{x}} \text{ and } p_{\overline{\boldsymbol{x}}}(\boldsymbol{x}_i) \neq p_{\overline{\boldsymbol{x}}}(\boldsymbol{x}_j) \text{ for any } i \neq j \in \mathcal{V}. \tag{3}$$

Point clouds satisfy this condition with probability 1 if they are uniformly distributed; see Lemma 3.4. We focus on this case in the main context and present a strategy in Appendix D to maintain the desired properties of our graph when the generic condition equation (3) does not hold.

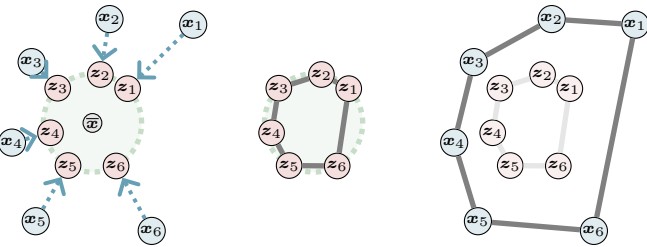

Figure 3: Visualizing the SCHull graph construction process: (left panel) The blue points $\boldsymbol{x}_i$s represent the given point cloud, and the gray node $\overline{\boldsymbol{x}}$ denotes the center of this point cloud. Each point is projected onto the green unit sphere centered at $\overline{\boldsymbol{x}}$, resulting in the corresponding orange points $\boldsymbol{z}_i = p_{\overline{\boldsymbol{x}}}(\boldsymbol{x}_i)$. (middle panel) The convex hull of the orange points $\boldsymbol{z}_i = p_{\overline{\boldsymbol{x}}}(\boldsymbol{x}_i)$ is constructed, with its edges highlighted in gray. (right panel) The gray edges on the convex hull of the projected points induce the graph structure on the original point cloud.

**Step 2: Construct the convex hull and SCHull graph.** Next, we construct a convex hull – using the QuickHull algorithm (Barber et al., 1996) – for the projected points $(\mathcal{V}, p_{\overline{\boldsymbol{x}}}(\boldsymbol{X}))$ obtained from Step 1, and we denote the resulting convex hull as $\text{Conv}\left(\{p_{\overline{\boldsymbol{x}}}(\boldsymbol{x}_i) \mid i \in \mathcal{V}\}\right)$. In particular, the boundary of this convex hull is a polyhedral graph with nodes, edges, and faces. By disregarding the face structure, we obtain a graph $(\mathcal{V}, \mathcal{E})$ that includes all points in $\mathcal{V}$ thanks to the generic condition equation (3). Our graph construction for the given point cloud $(\mathcal{V}, \boldsymbol{X})$ is then defined as $\mathcal{G} = (\mathcal{V}, \mathcal{E}, \boldsymbol{X})$. This graph includes the original point cloud $(\mathcal{V}, \boldsymbol{X})$, with edges defined by those on the convex hull of $(\mathcal{V}, p_{\overline{\boldsymbol{x}}}(\boldsymbol{X}))$. Specifically, nodes $i, j$ are connected by an edge in $\mathcal{E}$ if and only if their projected points on the unit sphere $p_{\overline{\boldsymbol{x}}}(\boldsymbol{x}_i), p_{\overline{\boldsymbol{x}}}(\boldsymbol{x}_j)$ are connected by an edge on the convex hull $\text{Conv}\left(\{p_{\overline{\boldsymbol{x}}}(\boldsymbol{x}_i) \mid i \in \mathcal{V}\}\right)$.

We demonstrate an example of applying our SCHull graph construction for a synthetic point cloud in Fig. 3. For simplicity, we assume all points $\{\boldsymbol{x}_i\}_{i=1}^6$ are coplanar. The green circle represents the unit sphere centered at the mean point $\overline{\boldsymbol{x}} = \frac{1}{6} \sum_{i=1}^6 \boldsymbol{x}_i$. Each $\boldsymbol{z}_i := p_{\overline{\boldsymbol{x}}}(\boldsymbol{x}_i)$ denotes the projection of $\boldsymbol{x}_i$ onto the green unit sphere. We then construct the convex hull $\text{Conv}\left(\{\boldsymbol{z}_i\}\right)$ of the projected points, with solid gray lines indicating the edges. Finally, the SCHull graph is created by connecting points $\boldsymbol{x}_i$ and $\boldsymbol{x}_j$ if their corresponding projections $\boldsymbol{z}_i$ and $\boldsymbol{z}_j$ are connected by an edge in the convex hull.

**Computational complexity.** To analyze the computational complexity of constructing SCHull graphs, let $m$ be the number of points in the point cloud. The overall cost is dominated by the convex hull construction, which utilizes QuickHull with a computational complexity of $\mathcal{O}(m \log m)$ (Barber et al., 1996). Other steps, including projecting points to the unit sphere and attributes computation (see Section 3.3), result in $\mathcal{O}(m)$ additional computational complexity.

## 3.2 CONNECTIVITY AND SPARSITY OF SCHULL GRAPHS

In this section, we prove that SCHull graphs are connected and sparse. We begin by presenting a fundamental proposition that outlines the properties of convex hulls formed by points on a sphere.

**Proposition 3.1.** *Let $\mathcal{Z} = \{z_j\}$ be a set of points on a sphere where $|\mathcal{Z}| \geq 3$ (the number of points is at least 3), and let $\mathrm{Conv}(\mathcal{Z})$ denote the convex hull of $\mathcal{Z}$ with its associated graph structure consisting of nodes, edges, and faces defined by its boundary. Then, we have*

1. *Connectivity: Any two points in $\mathcal{Z}$ are connected by a sequence of edges in $\mathrm{Conv}(\mathcal{Z})$.*

2. *Sparsity: The number of edges in $\mathrm{Conv}(\mathcal{Z})$ is no greater than $3|\mathcal{Z}| - 6$.*

*Moreover, $\mathrm{Conv}(\mathcal{Z})$ is a strictly convex geometric polyhedral graph if $z_j$s are not coplanar.*

Proposition 3.1 ensures that the convex hull of a finite set of points on a sphere is always connected, eliminating the possibility of isolated points. Additionally, the number of edges is bounded above by three times the number of nodes, ensuring sparsity. Notice the SCHull graph directly inherits the graph structure from the convex hull of the projected points on the unit sphere, along with its connectivity and sparsity. We summarize these implications in the following corollary:

**Corollary 3.2.** *For any point cloud $(\mathcal{V}, \boldsymbol{X})$ with more than 2 points and satisfying the generic condition equation (3), the SCHull graph $(\mathcal{V}, \mathcal{E}, \boldsymbol{X})$ is a connected geometric polyhedral graph and has a linear relationship between the number of edges and the number of nodes. Specifically, the edge-to-node ratio is bounded above by 3, i.e. $\frac{|\mathcal{E}|}{|\mathcal{V}|} < 3$.*

The theoretical results in Corollary 3.2 are numerically substantiated by the results in Fig. 1. In particular, SCHull graphs are always connected, and the ratios between the number of edges and the number of nodes are always bounded above by 3 for all molecules in the benchmark Fold dataset.

## 3.3 RIGIDITY OF SCHULL GRAPHS AND ITS ROLE IN GNN'S PERFORMANCE

SCHull graphs guarantee both sparsity and connectivity. However, a crucial question remains:

> *How do SCHull graphs impact the performance of GNNs?*

To answer this affirmatively, we delve into the rigidity of SCHull graphs; see Appendix B for an overview of rigidity. We first introduce a common assumption about the genericity of point clouds in graph rigidity studies; see e.g., (Laman, 1970; Asimow & Roth, 1978; Connelly, 2005). We then propose a simple attribute-based design for SCHull graphs and provide a theoretical analysis demonstrating that our SCHull graphs – with their geometric attributes on both edges and nodes – can effectively enable GNNs to distinguish between any two non-isomorphic generic point clouds.

**Definition 3.3** (Genericity of point clouds). A point cloud $(\mathcal{V}, \boldsymbol{X})$ is said to be *generic*[2] if

$$\text{No multivariate polynomial } P \text{ with rational coefficients satisfies } P(\boldsymbol{x}_1, \ldots, \boldsymbol{x}_m) = 0. \quad (4)$$

The genericity of a point cloud means that the points are arranged in a way that cannot be described by a simple polynomial equation with rational coefficients. It's important to note that almost all point clouds are generic, with exceptions forming a set of Lebesgue measure zero (Mityagin, 2015). In particular, the genericity of point clouds presented in Definition 3.3 is stronger than the generic condition equation (3) that we have assumed in Section 3.1.

**Lemma 3.4.** *Any generic point cloud satisfies the generic assumption equation (3).*

Moreover, generic point clouds also avoid the cases where all the projected points $\{p_{\overline{\boldsymbol{x}}}(\boldsymbol{x}_i)\}$ are coplanar; this leads to the following result, allowing us to apply Theorem 2.1.

**Lemma 3.5.** *Let $(\mathcal{V}, \boldsymbol{X})$ be a generic point cloud. Then $\mathrm{Conv}\left(\{p_{\overline{\boldsymbol{x}}}(\boldsymbol{x}_i) \mid i \in \mathcal{V}\}\right)$ – the convex hull on the sphere we constructed in Section 3.1 – is a strictly convex geometric polyhedral graph.*

According to Lemma 3.5 and Theorem 2.1, edge lengths $d_{ij}$ and dihedral angles $\tau_{ij}$ are sufficient to uniquely determine the spatial arrangement of $\mathrm{Conv}\left(\{p_{\overline{\boldsymbol{x}}}(\boldsymbol{x}_i) \mid i \in \mathcal{V}\}\right)$ due to its convexity. While the SCHull graph is also a geometric polyhedral graph, its lack of convexity raises concerns about whether edge lengths and dihedral angles are sufficient to determine the spatial arrangement of points.

---

[2]Also known as algebraically independent.

However, by including the original distances of the nodes from the center of the point clouds as node attributes, we can prove that any maximally expressive GNN – a standard framework for analyzing GNNs' performance – can distinguish between any two non-isomorphic point clouds (see Section 2 for a brief overview). Specifically, for a given point cloud $(\mathcal{V}, \boldsymbol{X})$, we consider the attributed version of our SCHull graph $\mathcal{G} = (\mathcal{V}, \mathcal{E}, \boldsymbol{X}, \boldsymbol{F})$, where we encode

$$
\begin{aligned}
&\text{the edge attributes } e_{ij} = (\|\boldsymbol{x}_i - \boldsymbol{x}_j\|, \tau_{ij}) \text{ for any } (i,j) \in \mathcal{E}, \text{ and} \\
&\text{the node attributes } f_i = \|\boldsymbol{x}_i - \overline{\boldsymbol{x}}\| \text{ for any } i \in \mathcal{V}.
\end{aligned}
\tag{5}
$$

Notice that $\tau_{ij}$ are the dihedral angles between faces computed on $\mathrm{Conv}\left(\{p_{\overline{\boldsymbol{x}}}(\boldsymbol{x}_i) \mid i \in \mathcal{V}\}\right)$. Under these settings, we have

**Theorem 3.6.** *Let $F$ be a maximally expressive GNN with depth $T = 1$. Then $F$ can distinguish between the attributed SCHull graphs defined in equation (5) of any two non-isomorphic generic point clouds.*

*Remark* 3.7. The node attribute design of SCHull in equation (5) can be omitted without affecting the theorem's result if we consider removing an additional measure zero subset of point clouds. Due to page limitations, this version of Theorem 3.6 is provided in Appendix F. Additionally, while we do not address the coplanar case in the main text, the corresponding theory and discussion regarding the effectiveness of our SCHull graph for coplanar cases can be found in Appendix E.

*Remark* 3.8. As discussed in the introduction, the rigidity property does not generally hold for other graph construction methods, e.g., radial cutoff. While Sverdlov & Dym (2024) propose power graphs as a solution to ensure rigidity in these methods, power graphs often lead to increased graph density and do not address cases where the original graph is disconnected. A more detailed introduction to power graphs is provided in Appendix A and an empirical study is provided in Appendix B. In contrast, the SCHull graph – when dealing with generic point clouds – naturally strikes a balance between rigidity and sparsity, making it an efficient and suitable option for GNN applications.

*Remark* 3.9. Existing works, such as (Joshi et al., 2023; Li et al., 2024; Hordan et al., 2024), have established theoretical frameworks that address both generic and non-generic point clouds. In particular, Li et al. (2024) highlights the challenges of distinguishing ambiguous symmetric point clouds. While our method does not currently provide a theoretical guarantee for such cases, we present empirical evidence in Appendix B showing that even shallow graph neural networks using our approach can effectively distinguish ambiguous symmetric point clouds – cases where incorporating all edge lengths on complete graphs fall short.

To solidify Theorem 3.6, we design a synthetic dataset named NestedSquares. The NestedSquares dataset comprises ten graphs, each consisting of two nested squares. In each graph, the inner square is rotated by an angle $\theta \in (0, \pi/2)$; see Fig. 4 for an example. Each graph is labeled with the angular rotation, and the objective is to minimize the mean squared error when predicting this angle. The data is split into 6 training graphs, 2 test

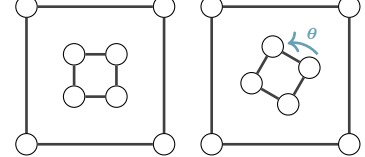

Figure 4: Depicting the rotation of the graph from the NestedSquares dataset. Ten rotated graphs are generated by randomly choosing $\theta$.

graphs, and 2 validation graphs. This task evaluates GNNs' ability to learn both local and global geometric information. To show the effectiveness of SCHull graphs, we compare message passing using solely the information provided by SCHull against existing geometric GNNs with common graph constructions. We introduce MPNN+SCHull, a message-passing neural network (MPNN) that takes the SCHull graph as input, incorporating both graph structure and the constructed edge attributes and node features as defined in equation (5); see Appendix G.2 for the detailed architecture of the used MPNN, and Appendix H.1 for experimental details.

|  | Radial cutoff | $k$NN ($k = 3$) | Voronoi | SCHull (ours) |
|---|---|---|---|---|
| MPNN | $0.657 \pm 0.165$ | $0.213 \pm 0.013$ | $0.281 \pm 0.049$ | $\mathbf{0.068 \pm 0.049}$ |
| DimeNet++ | $0.281 \pm 0.049$ | $0.281 \pm 0.049$ | $0.292 \pm 0.018$ | — |
| SphereNet | $0.661 \pm 0.212$ | $0.450 \pm 0.121$ | $0.494 \pm 0.128$ | — |
| ComENet | $0.885 \pm 0.287$ | $0.502 \pm 0.1928$ | $0.816 \pm 0.275$ | — |

Table 2: Test mean squared error (MSE) of single-layer invariant GNNs for the NestedShapes dataset. MPNN+SCHull utilizes the SCHull graph structure. Our method outperforms all other models and graph constructions. The mean $\pm$ standard deviation is reported from 10 random trials.

In particular, we cover several representative invariant GNN models, including DimeNet++ (Gasteiger et al., 2020a), SphereNet (Liu et al., 2022), and ComENet (Wang et al., 2022b). These invariant GNNs,

along with the simple MPNN, were equipped with various graphs, including Radial cutoff, $k$NN, and Voronoi. Notably, we did not implement other invariant GNNs on SCHull to avoid any potential biases introduced by their specific designs. Table 2 shows the advantage of SCHull over other graphs with different invariant GNN models. Notice that MPNN+SCHull significantly outperforms all other models almost by an order of magnitude.

### 3.4 SOME FURTHER DISCUSSIONS

While SCHull graphs have connectivity, sparsity, and rigidity guarantees, their performance in practice may vary depending on the complexity of the problem. Our proposed SCHull graph is more likely to connect distant nodes, capturing global information but potentially overlooking local details. To fully leverage the strengths of SCHull graphs, we recommend using them in conjunction with an additional informative sparse graph, such as a radial cutoff graph with a small cutoff threshold or a chemical graph. This combined approach can ensure both global geometric completeness and capture of local information while maintaining connectivity and sparsity.

To demonstrate the integration of SCHull graphs with existing GNNs that use radial cutoff graphs, we consider a geometric graph $\mathcal{G} = (\mathcal{V}, \mathcal{E}, \boldsymbol{X}, \boldsymbol{F})$ constructed using the radial cutoff method from the point cloud $(\mathcal{V}, \boldsymbol{X})$. We then construct the corresponding SCHull graph $\tilde{\mathcal{G}} = (\mathcal{V}, \tilde{\mathcal{E}}, \boldsymbol{X}, \tilde{\boldsymbol{F}})$. Notice that most GNNs – e.g., (Wang et al., 2022a; Liu et al., 2022; Jing et al., 2020) – utilize a predesigned embedding function $\text{EMB}(\mathcal{E}, \boldsymbol{X}, \boldsymbol{F})$, to obtain the edge attributes $\boldsymbol{e} = \{e_{ij} \mid (i, j) \in \mathcal{E}\}$ and the embedded node features $\boldsymbol{F}' = [\boldsymbol{f}'_1, \ldots, \boldsymbol{f}'_m]$ on the graph $\mathcal{G}$. These attributed graphs are then input into the GNN model. We apply the same embedding function to compute the edge attributes $\tilde{\boldsymbol{e}} = \{\tilde{e}_{ij} \mid (i, j) \in \tilde{\mathcal{E}}\}$ and the embedded node features $\tilde{\boldsymbol{F}}' = [\tilde{\boldsymbol{f}}'_1, \ldots, \tilde{\boldsymbol{f}}'_m]$ for the SCHull graph $\tilde{\mathcal{G}}$. It is worth mentioning that the embedded features $\tilde{\boldsymbol{e}}$ and $\tilde{\boldsymbol{F}}'$ contain the features mentioned in equation (5). We then define the integrated node features as $\boldsymbol{F}^* = \boldsymbol{F}' \oplus \tilde{\boldsymbol{F}}'$, and the integrated edges and edge attributes as $\mathcal{E}^* = \mathcal{E} \cup \tilde{\mathcal{E}}$ and $\boldsymbol{e}^* = \boldsymbol{e} \cup \tilde{\boldsymbol{e}}$, respectively. Finally, we obtain the SCHull integrated graph $\mathcal{G}^* = (\mathcal{V}, \mathcal{E}^*, \boldsymbol{X}, \boldsymbol{F}^*)$, which serves as the input to the GNNs.

## 4 NUMERICAL RESULTS

We present comprehensive empirical results to show the effectiveness of SCHull-integrated GNNs. Across various tasks, we highlight that: (1) The SCHull graph's connectivity and sparsity properties effectively capture geometric information while maintaining computational efficiency. (2) The enriched graph structure, coupled with comprehensive node and edge features, significantly enhances message-passing capabilities. To validate our approach, we integrate SCHull into several GNN models and evaluate their performance on both small- and large-scale graph tasks, including atom force prediction, protein fold and enzyme reaction classification, and protein property prediction. The results consistently demonstrate that SCHull-integrated GNN models outperform the baseline models across multiple metrics while maintaining computational efficiency.

**Software and equipment:** Our implementation relies on the PyTorch Geometric (Fey & Lenssen, 2019) and SciPy (Virtanen et al., 2020) frameworks. Experiments are conducted on a single NVIDIA RTX 3090 GPU, and T4 and A100 GPUs provided by Google Colab (Google Colaboratory, 2023).

**Training setup:** The training setups for different tasks are available in Appendices I.2 and I.3.

We integrate SCHull graphs into DimeNet (Gasteiger et al., 2020b), SphereNet (Liu et al., 2022) and LEFTNet (Du et al., 2024) for learning node-level equivariant features on molecule dataset MD17 (Chmiela et al., 2017). Also, we integrate SCHull into ProNet-Backbone, ProNet-Amino-Acid (Wang et al., 2022a), and GVP-GNN (Jing et al., 2020) for benchmark tasks of protein fold classification (Fold) and enzyme reaction classification (React) (Hou et al., 2018), and protein-ligand binding affinity prediction (LBA) (Wang et al., 2004; Liu et al., 2015).

**Metrics:** For prediction, we use various metrics depending on the tasks e.g., mean absolute error (MAE), root mean square error (RMSE), Pearson Correlation (Uebersax, 1987), Spearman Correlation (Panda & Pati, 2015) and Kendall Correlation (Bradley, 1976). For each task, we run the experiment five times and record the standard deviation of the corresponding metric.

### 4.1 PREDICTING ATOMIC FORCES FOR MD17 SMALL MOLECULES

In this experiment, we evaluate how effectively the SCHull framework enhances different models in learning equivariant features by predicting atomic forces on the MD17 dataset. We integrate

SCHull into LEFTNet, DimeNet, and SphereNet. Following prior research (Schütt et al., 2018; Liu et al., 2022; Du et al., 2024), we train individual models for each of the seven molecules: Aspirin, Benzene, Ethanol, Malonaldehyde, Naphthalene, Toluene, and Uracil. Detailed experimental and hyperparameter settings are provided in Appendices H.2 and I.2, respectively. Both the training and validation sets consist of 1,000 samples each, with the remaining data reserved for testing.

We use MAE to assess model performance on the test dataset. The results in Table 3 show that all SCHull-integrated models outperform baselines on MD17 while maintaining comparable runtime. This indicates that SCHull enhances the models' ability to capture equivariant information without significantly affecting message-passing efficiency.

| Molecule | DimeNet | DimeNet+SCHull | SphereNet | SphereNet+SCHull | LEFTNet | LEFTNet+SCHull |
|---|---|---|---|---|---|---|
| Aspirin | 0.499 | $0.427_{\pm.004}$ | 0.430 | $0.387_{\pm.005}$ | 0.281 | $\mathbf{0.240}_{\pm.005}$ |
| Benzene | 0.187 | $0.157_{\pm.006}$ | 0.178 | $0.155_{\pm.004}$ | 0.147 | $\mathbf{0.098}_{\pm.002}$ |
| Ethanol | 0.230 | $0.198_{\pm.003}$ | 0.208 | $0.181_{\pm.003}$ | 0.138 | $\mathbf{0.109}_{\pm.002}$ |
| Malonaldehyde | 0.383 | $0.334_{\pm.003}$ | 0.340 | $0.298_{\pm.003}$ | 0.205 | $\mathbf{0.151}_{\pm.002}$ |
| Naphthalene | 0.215 | $0.178_{\pm.002}$ | 0.178 | $0.144_{\pm.002}$ | 0.074 | $\mathbf{0.058}_{\pm.001}$ |
| Toluene | 0.210 | $0.169_{\pm.002}$ | 0.155 | $0.129_{\pm.002}$ | 0.083 | $\mathbf{0.076}_{\pm.001}$ |
| Uracil | 0.301 | $0.288_{\pm.002}$ | 0.267 | $0.242_{\pm.003}$ | 0.117 | $\mathbf{0.095}_{\pm.001}$ |
| Training Time/Epoch(s) | $43_{\pm0.9}$ | $50_{\pm0.8}$ | $51_{\pm1.0}$ | $62_{\pm1.5}$ | $24_{\pm0.5}$ | $28_{\pm0.5}$ |

Table 3: Test MAEs of MD17 dataset vector-valued properties prediction.

## 4.2 LEARNING PROTEINS

We demonstrate the importance of ensuring connectivity and sparsity in large-scale graph tasks, such as protein classification and property prediction. Protein graphs can contain thousands of nodes, making it crucial to maintain graph sparsity for computational efficiency while preserving connectivity to allow GNNs to capture global geometric information. We integrate SCHull into protein-task-specific GNNs – such as ProNet-Backbone, ProNet-Amino-Acid, GVP-GNN, SEGNN, and MACE – and evaluate them on three protein datasets. Details of the model training and hyperparameters are provided in Appendix I.3. The experimental results highlight the significance of capturing global information during message passing using SCHull features, particularly in large graphs. Additionally, including SCHull does not introduce significant overhead in terms of runtime.

### 4.2.1 FOLD CLASSIFICATION

Protein fold classification (Levitt & Chothia, 1976) informs the connection between protein structure and function, as well as protein evolution. In this experiment, we assess our SCHull integrated models for the protein fold classification task by following the fold dataset and experimental frameworks in Wang et al. (2022a). To evaluate generalization performance, three test sets are used:

- **Family:** includes proteins from the same family in the training data.
- **Superfamily:** excludes proteins from the same family appearing in the training data.
- **Fold:** excludes proteins from the same superfamily appearing in the training data.

More dataset information and splits can be found in Appendix H.3. Table 4 demonstrates that SCHull improves model accuracy across all three test datasets, with only a marginal increase in runtime. Furthermore, to assess how SCHull aids models in capturing global information, we divide each test set into four subsets based on the number of nodes (with at most 150, 300, 450, and 600 nodes). Fig. 5 shows that as graph size increases, the SCHull-integrated ProNet-Backbone achieves progressively higher accuracy over its original model. This result supports our claim that preserving connectivity is crucial for GNNs to learn global information and predict the class of protein data. Indeed, our proposed SCHull graph ensures connectivity, as shown in Fig. 1.

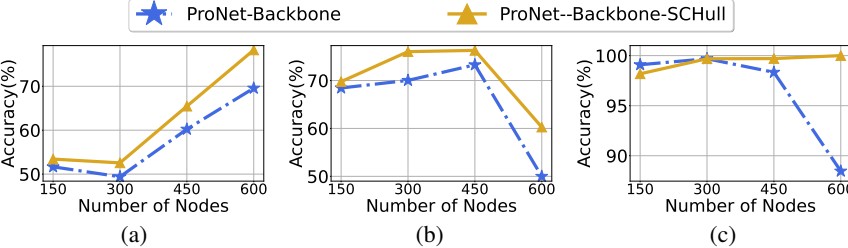

Figure 5: Comparison of the accuracy (%) on different sizes of graphs in the Fold classification dataset using ProNet(Backbon) and ProNet(Backbon)+SCHull. (a) Accuracy on test(Fold) dataset; (b) Accuracy on test(Super) dataset; (c) Accuracy on test(Family) dataset with at most 150, 300, 450, and 600 nodes.

| Method | React | Avg. Time | Fold | | | | Avg. Time |
| | | | Fold | Super | Family | Avg. | |
|---|---|---|---|---|---|---|---|
| GCN (Kipf & Welling, 2017) | 67.3 | | 16.8 | 21.3 | 82.8 | 40.3 | – |
| IEConv (Hermosilla et al., 2020) | 87.2 | – | 45.0 | 69.7 | 98.9 | 71.2 | – |
| DWNN (Li, 2022) | 76.7 | – | 31.8 | 37.8 | 85.2 | 51.5 | – |
| GearNet (Zhang et al., 2022) | 79.4 | – | 28.4 | 42.6 | 95.3 | 55.4 | – |
| HoloProt (Somnath et al., 2021) | 78.9 | – | – | – | – | – | – |
| MACE (Batatia et al., 2022) | – | – | $23.7_{\pm 0.5}$ | $21.4_{\pm 0.5}$ | $60.2_{\pm 0.2}$ | 35.1 | $114_{\pm 0.5}$ |
| MACE+SCHull | – | – | $27.0_{\pm 0.6}$ | $23.1_{\pm 0.5}$ | $65.0_{\pm 0.2}$ | 38.4 | $135_{\pm 0.5}$ |
| SEGNN (Brandstetter et al.) | – | – | $28.8_{\pm 0.6}$ | $30.3_{\pm 0.6}$ | $77.1_{\pm 0.3}$ | 45.4 | $121_{\pm 0.7}$ |
| SEGNN+SCHull | – | – | $32.0_{\pm 0.4}$ | $36.8_{\pm 0.7}$ | $86.9_{\pm 0.3}$ | 51.9 | $152_{\pm 0.5}$ |
| GVP-GNN (Jing et al., 2020) | 65.5 | $320_{\pm 5}$ | 16.0 | 22.5 | 83.8 | 40.8 | $106.3_{\pm 0.5}$ |
| **GVP-GNN + SCHull** | $77.1_{\pm 0.5}$ | $345_{\pm 5}$ | $24.5_{\pm 0.3}$ | $27.1_{\pm 0.2}$ | $88.6_{\pm 0.3}$ | 46.7 | $111.5_{\pm 0.5}$ |
| ProNet-Amino-Acid (Wang et al., 2022a) | 86.0 | $210_{\pm 5}$ | 51.5 | 69.9 | 99.0 | 73.5 | $70.5_{\pm 0.5}$ |
| **ProNet-Amino Acid+SCHull** | $87.9_{\pm 0.3}$ | $221_{\pm 6}$ | $55.2_{\pm 0.2}$ | $73.9_{\pm 0.2}$ | $99.1_{\pm 0.1}$ | 76.1 | $73.8_{\pm 0.5}$ |
| ProNet-Backbone (Wang et al., 2022a) | 86.4 | $213_{\pm 5}$ | 52.7 | 70.3 | 99.3 | 74.1 | $71.4_{\pm 0.8}$ |
| **ProNet-Backbone+SCHull** | $88.1_{\pm 0.3}$ | $230_{\pm 5}$ | $56.1_{\pm 0.3}$ | $74.6_{\pm 0.2}$ | $99.4_{\pm 0.1}$ | 76.7 | $75.8_{\pm 0.5}$ |

Table 4: Accuracy (%) on protein fold and enzyme reaction classification tasks. **Ave. Time** denotes the average time per training epoch. The top results are in boldface. SCHull consistently improves baseline models.

### 4.2.2 REACTION CLASSIFICATION

Enzymes are proteins functioning as biological catalysts. They can be categorized using enzyme commission (EC) numbers which classify enzymes according to the reactions they facilitate (Omelchenko et al., 2010). In this experiment, we evaluate the SCHull-integrated models on the reaction classification task using the same reaction dataset and experimental settings as outlined in (Wang et al., 2022a; Hou et al., 2018). Additional details on the dataset and the training, validation, and test splits are provided in Appendix H.3. Similar to the fold classification results in Section 4.2.1, Table 4 shows that SCHull consistently outperforms the original models, with a minimal increase in runtime.

### 4.2.3 LIGAND BINDING AFFINITY

Predicting protein-ligand binding affinity (LBA) plays a crucial role in various downstream processes in drug discovery. For this task, we use the SCHull-integrated models to predict LBA. The dataset is sourced from PDBbind (Wang et al., 2004; Liu et al., 2015) along with the experimental protocols established by (Jing et al., 2020). We employ default dataset split. See Appendix H.3 for more details of the dataset. We use a variety of statistical metrics, including RMSE, Pearson, Spearman, and Kendall correlations, to evaluate how SCHull enhances the learning capacity and generalization of GNNs. Again, Table 5 shows that SCHull-integrated models consistently outperform the original models across different metrics while maintaining high computational efficiency.

| Method | LBA | | | | Avg. Time |
| | RMSE↓ | Pearson↑ | Spearman↑ | Kendall↑ | |
|---|---|---|---|---|---|
| IEConv (Hermosilla et al., 2020) | 1.554 | 0.414 | 0.428 | – | – |
| HoloProt-Full Surface (Somnath et al., 2021) | 1.464 | 0.509 | 0.500 | – | – |
| HoloProt-Superpixel (Somnath et al., 2021) | 1.491 | 0.491 | 0.482 | – | – |
| GVP-GNN (Jing et al., 2020) | $1.529_{\pm 0.001}$ | $0.441_{\pm 0.001}$ | $0.432_{\pm 0.002}$ | $0.301_{\pm 0.002}$ | $48.6_{\pm 0.6}$ |
| **GVP-GNN + SCHull** | $1.401_{\pm 0.001}$ | $0.475_{\pm 0.001}$ | $0.459_{\pm 0.001}$ | $0.335_{\pm 0.002}$ | $53.6_{\pm 0.6}$ |
| ProNet-Amino-Acid (Wang et al., 2022a) | 1.455 | 0.536 | 0.526 | $0.465_{\pm 0.001}$ | $31.7_{\pm 0.5}$ |
| **ProNet-Amino Acid+SCHull** | $1.355_{\pm 0.002}$ | $0.556_{\pm 0.001}$ | $0.568_{\pm 0.001}$ | $0.512_{\pm 0.001}$ | $33.9_{\pm 0.5}$ |
| ProNet-Backbone (Wang et al., 2022a) | 1.458 | 0.546 | 0.550 | $0.481_{\pm 0.001}$ | $32.1_{\pm 0.5}$ |
| **ProNet-Backbone+SCHull** | $1.321_{\pm 0.002}$ | $0.581_{\pm 0.001}$ | $0.578_{\pm 0.1}$ | $0.535_{\pm 0.001}$ | $34.4_{\pm 0.5}$ |

Table 5: RMSE/Pearson Correlation/Spearman Correlation/Kendall Correlation on the LBA Test Dataset. **Ave. Time** refers to the average running time of one epoch in model training.

## 5 CONCLUDING REMARKS

In this paper, we introduce SCHull – a new computationally efficient and scalable method for constructing graph representations of molecules, large proteins, and general data with 3D point clouds. The SCHull graphs are guaranteed to be sparse, connected, and rigid; these properties are essential for learning biomolecules using graph neural networks. Furthermore, our proposed SCHull graph can be seamlessly integrated into the existing learning framework to boost the performance of existing models. There are a few avenues for future work. An important avenue is investigating whether the SCHull graph can ensure the maximal expressive power of GNNs for distinguishing non-generic point clouds, as demonstrated in the experiments in Appendix B.

## 6 Acknowledgement

This material is based on research sponsored by NSF grants DMS-2152762, DMS-2208361, DMS-2219956, and DMS-2436344, and DOE grants DE-SC0023490, DE-SC0025589, and DE-SC0025801. Moreover, this work is also partially supported by the U.S. Department of Energy Advanced Scientific Computing Research (ASCR) under DOE-FOA-2493 "Data-intensive scientific machine learning".

## 7 Ethics Statement

In this paper, we propose a new graph representation for geometric data with 3D coordinates, especially molecules. The new graph solves some fundamental challenges of existing graph representations for molecules. In particular, existing graphs struggle to ensure sparsity, connectivity, and rigidity. These merits are essential for the success of machine learning for molecular modeling. Our work belongs to fundamental research and are expect to improve existing models for learning geometric data. Our work is methodological, and we validate our proposed approaches on the benchmark datasets. We do not expect to cause negative societal problems. Furthermore, we do not see any issues with potential conflicts of interest and sponsorship, discrimination/bias/fairness concerns, privacy and security issues, legal compliance, and research integrity issues (e.g., IRB, documentation, research ethics.

## 8 Reproducibility Statement

We are committed to conducting reproducible research. To ensure the integrity and transparency of our work, we employ a multifaceted approach: First, we meticulously compare the novelty of our research against existing literature. This involves a thorough examination of the current state of the field to identify gaps in knowledge and demonstrate the unique contributions of our work. Second, we provide detailed derivations of our proposed approaches and theoretical results. By carefully outlining the mathematical underpinnings of our methods, we enhance the understanding of our work and facilitate its verification by others. Third, we conduct rigorous experiments using widely recognized benchmark datasets. This allows us to evaluate the performance of our methods against established standards and provides a solid foundation for comparison with other approaches. Fourth, we meticulously report experimental details, including the specific datasets used, parameters chosen, and evaluation metrics employed. Finally, we make all experimental codes, accompanied by comprehensive documentation, publicly available. This open-source approach empowers researchers to inspect our methods, verify our results, and build upon our work. By sharing our code, we foster collaboration, advance the field, and contribute to the overall reproducibility of scientific research.

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

## A ADDITIONAL RELATED WORKS.

**Invariant graph neural networks and completeness of geometric attributes.**   Invariant graph neural networks (GNNs) have emerged as powerful tools for analyzing geometric graphs by leveraging invariant geometric features (attributes) that remain unchanged under transformations like rotation and translation. The effectiveness of invariant GNNs hinges on the development of such features that not only remain invariant but also truly capture the essential geometric information carried by the graph. Several studies have explored methods for learning different meaningful invariant geometric features that effectively capture the underlying geometric structure of the data. SchNet (Schütt et al., 2018) utilizes pairwise distances between nodes as basic invariant geometric attributes. SphereNet (Liu et al., 2022) employs a local spherical coordinate system around each node to generate invariant distance and angular features. GemNet (Gasteiger et al., 2021) and ComENet (Wang et al., 2022b) incorporate torsion angles computed on node quadruplets, which is particularly useful for predicting the properties of molecular conformers. Similar to the way torsion angles capture the angles between local structures, LEFTNet (Du et al., 2024) leverages local frames centered on each node, allowing them to scalarize all spatial information into a set of invariant features and then combine them with frame transition information to capture global features.

Notably, Wang et al. (2022b) introduced the notion of completeness of geometric attributes on the graph, which is closely related to rigidity. Geometric attributes on the graph are considered complete if they can uniquely determine the spatial arrangement of the nodes within the graph. In fact, Wang et al. (2022b) demonstrated that their proposed framework ComENet is complete. However, as noted in (Wang et al., 2024), most existing methods, including all the models mentioned above, rely on fixed-size neighborhoods, which may be insufficient to capture all relevant global geometric features. While our graph rigidity proof hinges on the assumption of generic point clouds, it's crucial to recognize that the graph structure, formed through a global assessment of geometric arrangement, cannot be inferred solely from local information.

**Power graphs.**   The $k^{\text{th}}$-power graph of a given graph connects any two nodes if there exists a path between them in the original graph with length $\le k$. Sverdlov & Dym (2024) propose $4^{\text{th}}$-power graphs as a potential solution to achieve rigidity in 3D geometric graphs. While power graphs may offer a potential approach to achieve rigidity in generic point clouds, they can lead to overly dense graphs, which can be inefficient for large-scale structures. Additionally, power graphs require the original graph to be connected for the resulting power graph to remain connected. Furthermore, they may fail to help GNNs distinguish between two ambiguous, non-generic point clouds. These limitations are demonstrated in Tables 6 and 7.

**High-order graphs.**   Several studies have explored incorporating higher-order structures into graphs to capture more detailed geometric information. Notable examples include simplicial complexes (Eijkelboom et al., 2023) and combinatorial complexes (Battiloro et al., 2024). While these approaches show promise for encoding and learning geometric properties, they often suffer from high computational complexity during their construction, making them less efficient than our approach, especially when dealing with large proteins, along with other potential limitations. Specifically, simplicial complexes, constructed in (Eijkelboom et al., 2023), rely on radius-based connectivity, which may not guarantee connectivity (and hence may not guarantee rigidity) in all cases, as shown in Fig. 1. However, it would be interesting to explore how our graph construction method could be used as a foundation for generating simplicial complexes and how these resulting complexes might perform in capturing geometric and structural properties. Combinatorial complexes provide a more general framework for higher-order structures, encompassing simplicial complexes as a subset. Of particular interest is the concept of molecular combinatorial complexes introduced in (Battiloro et al., 2024). These complexes incorporate atoms, bonds, rings, and functional groups as cells of complexes. Developing invariant features to ensure the rigidity of these complexes presents an exciting avenue for future research. To the best of our knowledge, however, no prior work has explored this direction. We believe it would be worthwhile to investigate whether sparse, connected, and rigid simplicial or combinatorial complexes can be effectively constructed and applied in higher-order models.

**Graph construction for symmetric detections**   The graph construction method we propose shares similarities with the one introduced in (Welzl et al., 1988) for symmetry detection, later adapted in the ML community to construct equivariant frames (Baker et al., 2024). However, our approach defines the graph on the original point clouds, rather than solely on the projected point clouds as in the referenced work. Furthermore, our design of graph attributes differs significantly: while Welzl

et al. (1988) uses interior angles of facets, we employ dihedral angles between facets as the key attribute in our graph construction.

## B  ADDITIONAL DISCUSSIONS

**Rigidity of graphs.**    Graph rigidity, in its traditional definition, examines the geometric constraints imposed on a graph by fixed edge lengths. A graph with fixed edge lengths is considered *rigid* if it allows only one possible realization of node positions, up to rigid motions; for a comprehensive understanding, we direct readers to (Asimow & Roth, 1978; Connelly, 2005). To illustrate this concept, consider a simple 2D convex graph with four nodes and four equal-length edges. As demonstrated in Fig. 6, this graph is not rigid; it can take different geometric forms, such as a square or a rhombus. To make this convex graph rigid in 2D space, we add a diagonal edge (see Fig. 7). This addition results in a structure where edge lengths uniquely determine node coordinates, up to isometry. However, rigidity can be generalized beyond edge lengths. Incorporating geometric invariants like interior or dihedral angles provides a broader perspective. For instance, fixing the four interior angles of our example graph also ensures rigidity as shown in Fig. 7. This extended notion, initially explored by Cauchy and further developed in recent studies such as Cho & Kim (2023), addresses the rigidity of nonconvex polyhedra with respect to both edge lengths and dihedral angles. Building upon this extended notion of rigidity, we establish theoretical guarantees for the performance of Schull graph-based graph neural networks on general point clouds in Section 3.3.

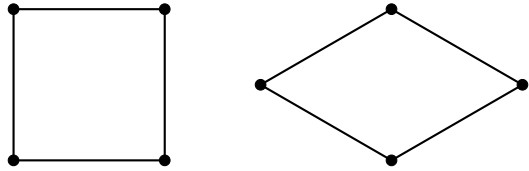

Figure 6: A four-node graph with four edges of equal fixed length, which can form either a square (left) or a rhombus (right), illustrating its non-rigidity.

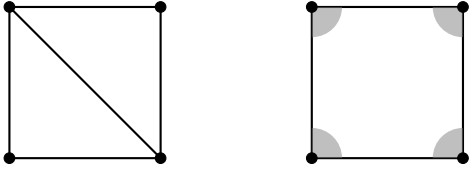

Figure 7: Two approaches to achieving graph rigidity: on the left, a square with a diagonal, demonstrating rigidity through fixed edge lengths; on the right, the same square with shaded angles, illustrating rigidity via fixed interior angles.

**Non-generic point clouds.**    Several existing works have explored the expressivity of GNNs on non-generic point clouds. For instance, $k$-DisGNNs (Li et al., 2024) aim to address the limitations of GNNs in fully leveraging the distance graph (i.e., a complete graph with distance as edge attributes) by designing higher-order GNN models that can better distinguish non-generic point clouds. Similarly, Hordan et al. (2024) propose WeLNet, which utilizes the distance graph and provides a theoretical guarantee that it can distinguish between all non-isometric 3D point clouds.

In this section, we apply our method to non-generic point clouds, particularly focusing on symmetric point clouds to assess if it improves traditional GNNs' ability to distinguish them. Prior work, such as (Li et al., 2024), has shown that even when distances are incorporated into a complete graph – where every pair of nodes is connected – GNNs can still struggle with special cases, such as collections of nodes from the vertices of convex regular polyhedrons.

To investigate whether incorporating our SCHull graph addresses these challenges, we test it using counterexamples of 6-point and 14-point configurations from (Li et al., 2024); see Fig. 8 for a visualization. Specifically, we adapt the EGNN architecture from (Satorras et al., 2021) by removing equivariant updates and replacing edge attributes with invariant features such as distances and dihedral angles. Experiments were conducted on a dataset consisting of augmented counterexample pairs, each subjected to

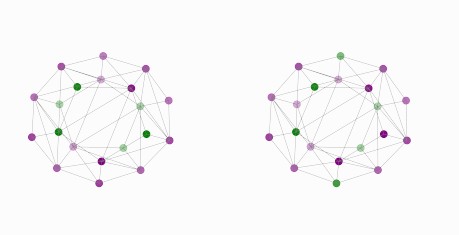

Figure 8: Visualization of pairs of 6-point (green) and 14-point (purple) configurations from (Li et al., 2024), where the edges represent the dodecahedron.

20 random rotations, over 100 epochs, with results averaged over 10 independent runs. We report the mean accuracy and standard deviation to ensure robustness. The empirical results in Tables 6 and 7 demonstrate that the edge attribute design of the SCHull graph (equation (5)) enables even shallow GNNs to distinguish between two non-isomorphic geometric graphs successfully.

To provide further validation and comparisons, we have also tested the efficacy of using only edge distances within the SCHull graph and evaluated alternative graph construction methods, including radial cutoff graphs with varying thresholds, power graphs (Sverdlov & Dym, 2024), and complete graphs (Li et al., 2024; Hordan et al., 2024). The graph properties and performance for all methods are summarized in Tables 6 and 7, with tuple notations representing distinct graph properties on paired point clouds. Additionally, visualizations of the graphs for the 14-point configurations are provided in Table 8, offering a clearer geometric perspective on the differences between the approaches.

| | 1 Layer | 2 Layers | # Edges / # Nodes | # Components |
|---|---|---|---|---|
| **Radius Graph ($r = 1.8$) w/ Distance** | $50.0 \pm 0.0$ | $50.0 \pm 0.0$ | 0.33 | 4 |
| **Radius Graph ($r = 2.5$) w/ Distance** | $50.0 \pm 0.0$ | $50.0 \pm 0.0$ | 1 | (2, 1) |
| **Radius Graph ($r = 3.0$) w/ Distance** | $50.0 \pm 0.0$ | $50.0 \pm 0.0$ | 1.67 | 1 |
| **$4^{\text{th}}$-power Radius Graph ($r = 1.8$) w/ Distance** | $50.0 \pm 0.0$ | $50.0 \pm 0.0$ | 0.33 | 4 |
| **$4^{\text{th}}$-power Radius Graph ($r = 2.5$) w/ Distance** | $\mathbf{100.0 \pm 0.0}$ | $\mathbf{100.0 \pm 0.0}$ | (1, 2.5) | (2, 1) |
| **$4^{\text{th}}$-power Radius Graph ($r = 3.0$) w/ Distance** | $50.0 \pm 0.0$ | $50.0 \pm 0.0$ | 2.5 | 1 |
| **Complete Graph w/ Distance** | $50.0 \pm 0.0$ | $50.0 \pm 0.0$ | 2.5 | 1 |
| **SCHull w/ Distance** | $50.0 \pm 0.0$ | $50.0 \pm 0.0$ | 2.0 | 1 |
| **SCHull w/ Distance and Dihedral Angles** | $\mathbf{100.0 \pm 0.0}$ | $\mathbf{100.0 \pm 0.0}$ | 2.0 | 1 |

Table 6: Comparison of graph properties and GNN performance (Unit:%) on 6-point symmetric point clouds using different graph construction methods. Tuple notations represent distinct graph properties on paired point clouds.

| | 1 Layer | 2 Layers | # Edges / # Nodes | # Components |
|---|---|---|---|---|
| **Radius Graph ($r = 1.8$) w/ $d_{ij}$** | $50.0 \pm 0.0$ | $50.0 \pm 0.0$ | 1.0 | (1, 2) |
| **Radius Graph ($r = 2.5$) w/ $d_{ij}$** | $50.0 \pm 0.0$ | $51.0 \pm 3.0$ | 3.0 | 1 |
| **Radius Graph ($r = 3.0$) w/ $d_{ij}$** | $60.5 \pm 11.1$ | $57.2 \pm 9.1$ | 5.0 | 1 |
| **$4^{\text{th}}$-power Radius Graph ($r = 1.8$) w/ $d_{ij}$** | $\mathbf{100.0 \pm 0.0}$ | $\mathbf{100.0 \pm 0.0}$ | (5, 3) | (1, 2) |
| **$4^{\text{th}}$-power Radius Graph ($r = 2.5$) w/ $d_{ij}$** | $50.0 \pm 0.0$ | $50.0 \pm 0.0$ | 6.5 | 1 |
| **$4^{\text{th}}$-power Radius Graph ($r = 3.0$) w/ $d_{ij}$** | $50.0 \pm 0.0$ | $50.0 \pm 0.0$ | 6.5 | 1 |
| **Complete Graph w/ $d_{ij}$** | $59.0 \pm 6.6.0$ | $50.0 \pm 0.0$ | 6.5 | 1 |
| **SCHull w/ $d_{ij}$** | $\mathbf{100.0 \pm 0.0}$ | $\mathbf{100.0 \pm 0.0}$ | 2.57 | 1 |
| **SCHull w/ $d_{ij}$ and $\tau_{ij}$** | $\mathbf{100.0 \pm 0.0}$ | $\mathbf{100.0 \pm 0.0}$ | 2.57 | 1 |

Table 7: Comparison of graph properties and GNN performance (Unit:%) on 14-point symmetric point clouds using different graph construction methods. Tuple notations represent distinct graph properties on paired point clouds.

The results showcase the limitations of alternative methods: radius graphs face issues of disconnectivity at small radii and edge redundancy at larger radii, neither of which ensures consistent performance; power graphs exhibit inconsistent performance that depends heavily on careful parameter tuning, and they fail to address issues of sparsity or connectivity effectively; complete graphs – while providing exhaustive connectivity – are computationally expensive and do not improve accuracy. In contrast, the SCHull graph achieves a balance of sparsity and rigidity, offering a compact yet expressive representation that enhances GNN performance. Its efficiency and scalability make it particularly appealing for large datasets, enabling effective learning with minimal computational overhead.

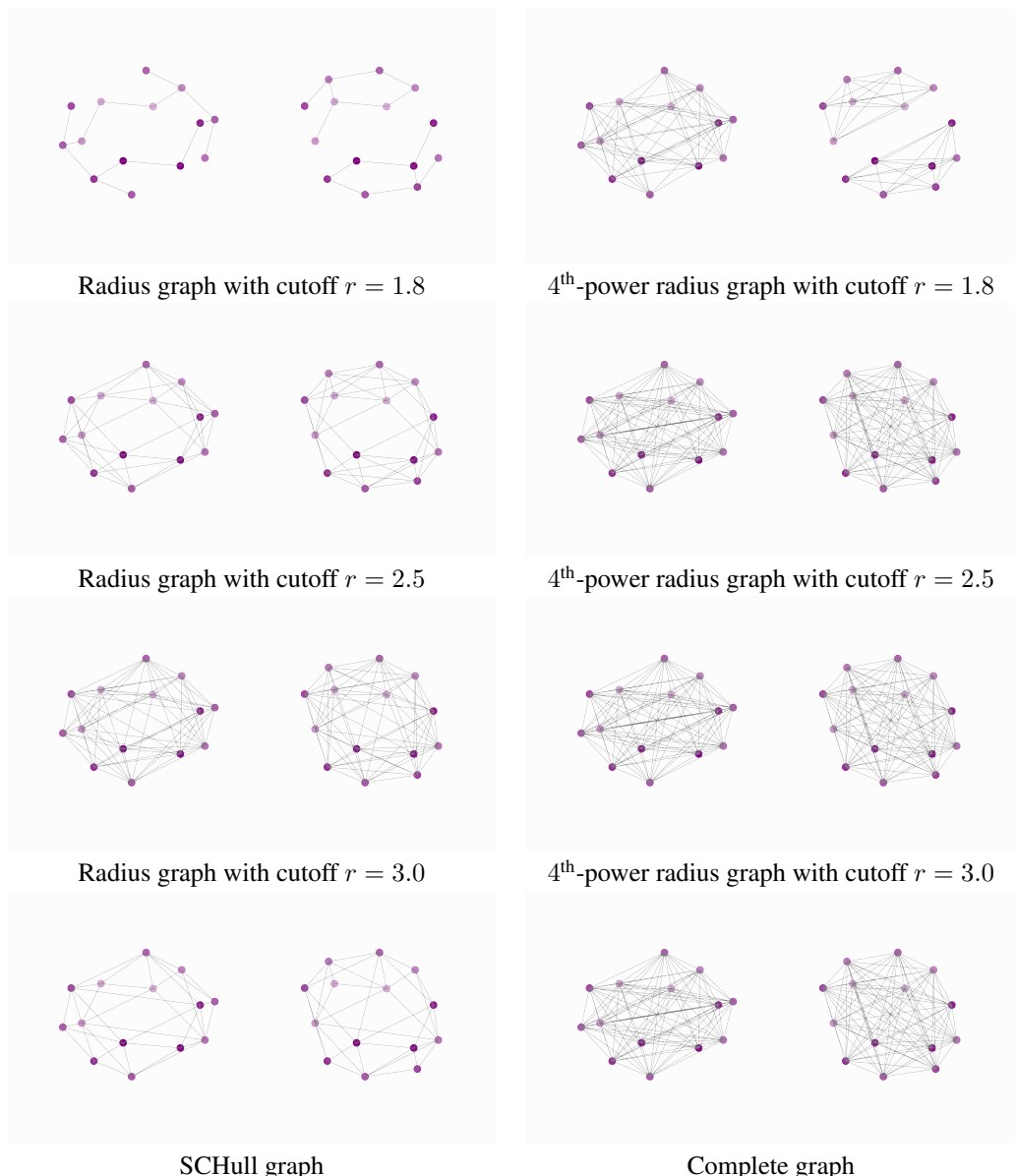

Table 8: We visualize the graphs of 14-points point clouds used in the experiments whose graph properties and performance with GNNs are presented in Table 7.

These findings emphasize the SCHull graph's robustness and efficiency in distinguishing symmetric point clouds, confirming its design as both expressive and computationally efficient. However, a formal theoretical analysis of this approach for non-generic point clouds is left as a direction for future work.

**Geometric Interpretation.** We present an intuitive explanation of the SCHull graph construction to clarify its connection to the convex hull and its advantages in representing point clouds.

The convex hull captures only the outermost boundary of a point cloud, omitting information about interior points. This is evident in Fig. 9, where the convex hull of $\{x_i\}_{i=1}^{6}$ (left) aligns with the SCHull graph for those points. However, for $\{x_i\}_{i=1}^{8}$ (right), the convex hull remains the same as for $\{x_i\}_{i=1}^{6}$, failing to account for interior points $x_7, x_8$. In contrast, SCHull connects interior points, preserving the structural relationships necessary to recover both the convex hull and the full geometry of the point cloud.

SCHull's design addresses this limitation by leveraging spherical projection. Points are mapped onto the unit sphere, where the graph structure is computed. This ensures that interior points contribute to the graph's structure. Conceptually, this transformation is akin to converting 3D Cartesian coordinates $(x, y, z)$ into spherical coordinates $(r, \theta, \phi)$, where the radial distance $r$ is excluded in the graph construction. The SCHull graph, therefore, relies only on the angular coordinates $(\theta, \phi)$, capturing the intrinsic geometry of the point cloud. One can see this rationale through the polar grid in Fig. 9. Importantly, the radial coordinate $r$ is retained as a node feature. This allows for the reconstruction of the original 3D coordinates $(r, \theta, \phi)$ and allows SCHull to represent global geometric properties effectively. For example, as shown in Fig. 9, edge lengths like $\|\boldsymbol{x}_1 - \boldsymbol{x}_7\|$ and $\|\boldsymbol{x}_1 - \boldsymbol{x}_6\|$ can be used together with node features $\|\boldsymbol{x}_1\|, \|\boldsymbol{x}_6\|, \|\boldsymbol{x}_7\|$ to compute information on the convex hull, such as $\|\boldsymbol{x}_1 - \boldsymbol{x}_6\|$.

In summary, SCHull preserves the global geometry of the convex hull while enriching it with the structural representation of interior points. This makes SCHull an effective tool for capturing the full spatial structure of point clouds and their intrinsic relationships. We note that Fleming & Fleming (2018) uses the convex hull to estimate hydrodynamic volume, and we believe that the SCHull graph could not only improve the prediction of this property but also offer additional insights.

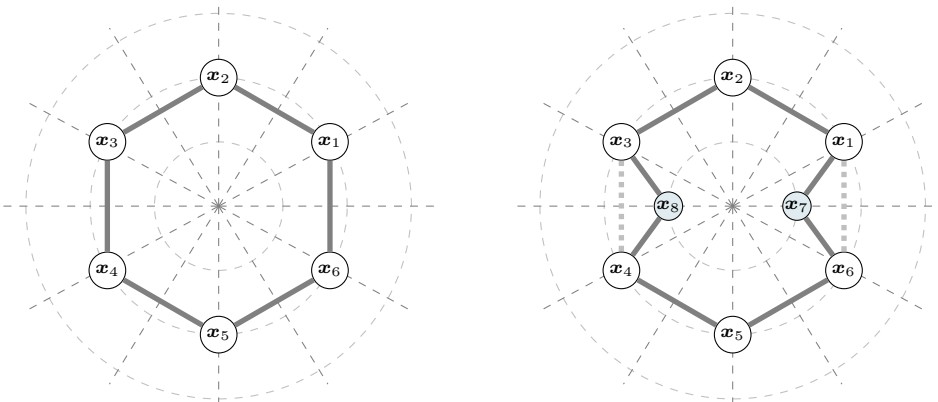

Figure 9: This figure highlights the close relationship between the convex hull of point clouds and SCHull graphs. While the convex hull captures only the outer boundary, potentially discarding information about interior points, the SCHull graph retains this information by connecting interior points. The edges of the SCHull graph are represented by solid gray lines, while the dashed gray lines indicate convex hull edges that differ from those of the SCHull graph. This distinction emphasizes SCHull's ability to preserve and enrich geometric relationships beyond the convex hull.

**Discontinuous intrinsic properties of graph construction.** The discrete nature of graph data inherently prevents the possibility of truly continuous graph construction, a limitation shared by all graph construction methods, including the one proposed in this work. While this was not the primary focus of our discussion, it raises an intriguing question: how does the choice of graph construction, especially those with attribute designs, affect the smoothness properties of end-to-end neural network models? Specifically, if the input data is perturbed by noise or errors, can we theoretically analyze how different graph construction methods influence the resulting impact on the neural network's output? For instance, does the output exhibit stability under input perturbations such that the end-to-end models are still continuous?

## C MISSING PROOFS AND ADDITIONAL DETAILS

**Proposition 3.1.** *Let $\mathcal{Z} = \{\boldsymbol{z}_j\}$ be a set of points on a sphere where $|\mathcal{Z}| \geq 3$ (the number of points is at least 3), and let $\mathrm{Conv}(\mathcal{Z})$ denote the convex hull of $\mathcal{Z}$ with its associated graph structure consisting of nodes, edges, and faces defined by its boundary. Then, we have*

    *1. Connectivity: Any two points in $\mathcal{Z}$ are connected by a sequence of edges in $\mathrm{Conv}(\mathcal{Z})$.*

    *2. Sparsity: The number of edges in $\mathrm{Conv}(\mathcal{Z})$ is no greater than $3|\mathcal{Z}| - 6$.*

*Moreover,* $\text{Conv}(\mathcal{Z})$ *is a strictly convex geometric polyhedral graph if* $\boldsymbol{z}_j$*s are not coplanar.*

*Proof of Proposition 3.1.* Since the point cloud $\mathcal{Z}$ lies on a sphere, each point in $\mathcal{Z}$ is an extreme point (Leonard & Lewis, 2015) of $\text{Conv}(\mathcal{Z})$. Consequently, each point appears as a node in the graph structure of the convex hull. If the points in $\mathcal{Z}$ are not coplanar, then $\text{Conv}(\mathcal{Z})$ is a convex polyhedron, and, hence by definition, $\text{Conv}(\mathcal{Z})$ is a strictly convex polyhedral graph. Since each convex polyhedron can be represented as a 3-connected planar graph by Steinitz's theorem, $\text{Conv}(\mathcal{Z})$ is connected, and the number of its edges is bounded above by $3|\mathcal{Z}| - 6$ (Lipton & Tarjan, 1979; O'Rourke, 1998). If the points in $\mathcal{Z}$ are coplanar, then $\text{Conv}(\mathcal{Z})$ itself is a connected planar graph. Therefore, the same bound of $3|\mathcal{Z}| - 6$ on the number of edges holds. $\square$

**Lemma 3.4.** *Any generic point cloud satisfies the generic assumption equation (3).*

*Proof of Lemma 3.4.* If $\boldsymbol{x}_i = \overline{\boldsymbol{x}}$ for some $i \in \mathcal{V}$, it directly defines a linear algebraic dependence on the coordinates, leading to a contradiction. Similarly, suppose $p_{\overline{\boldsymbol{x}}}(\boldsymbol{x}_i) = p_{\overline{\boldsymbol{x}}}(\boldsymbol{x}_j)$ for some $i, j \in \mathcal{V}$. Then we have $\frac{(\boldsymbol{x}_i - \overline{\boldsymbol{x}})}{\|(\boldsymbol{x}_i - \overline{\boldsymbol{x}})\|} = \frac{(\boldsymbol{x}_j - \overline{\boldsymbol{x}})}{\|(\boldsymbol{x}_j - \overline{\boldsymbol{x}})\|}$, implying that $(\boldsymbol{x}_i - \overline{\boldsymbol{x}}) \cdot (\boldsymbol{x}_j - \overline{\boldsymbol{x}}) = \|\boldsymbol{x}_i - \overline{\boldsymbol{x}}\| \|\boldsymbol{x}_j - \overline{\boldsymbol{x}}\|$. Taking the square of both sides, we obtain an algebraic dependence on the coordinates, which leads to a contradiction. $\square$

**Lemma 3.5.** *Let* $(\mathcal{V}, \boldsymbol{X})$ *be a generic point cloud. Then* $\text{Conv}(\{p_{\overline{\boldsymbol{x}}}(\boldsymbol{x}_i) \mid i \in \mathcal{V}\})$ *– the convex hull on the sphere we constructed in Section 3.1 – is a strictly convex geometric polyhedral graph.*

*Proof of Lemma 3.5.* Due to Proposition 3.1, it suffices to show that $\{p_{\overline{\boldsymbol{x}}}(\boldsymbol{x}_i) \mid i \in \mathcal{V}\}$ are not coplanar. This is true unless the original point cloud lies on a plane, which contradicts the genericity assumption. $\square$

**Theorem 3.6.** *Let* $F$ *be a maximally expressive GNN with depth* $T = 1$*. Then* $F$ *can distinguish between the attributed SCHull graphs defined in equation (5) of any two non-isomorphic generic point clouds.*

*Proof of Theorem 3.6.* The proof is inspired by the work Sverdlov & Dym (2024). Consider two SCHull graphs, $\mathcal{G}$ and $\mathcal{G}'$, generated from point clouds $(\mathcal{V}, \boldsymbol{X})$ and $(\mathcal{V}', \boldsymbol{X}')$, respectively. If the maximally expressive GNN $F$ assigns the same global feature to $\mathcal{G}$ and $\mathcal{G}'$ after one iteration, then the multisets of node features $\{\!\{s_i\}\!\}$ and $\{\!\{s_i'\}\!\}$ are identical. This implies $|\mathcal{V}| = |\mathcal{V}'|$. By relabeling, if necessary, we can assume that $s_i = s_i'$ for all $i \in \mathcal{V}$. This further implies that for each node $i \in \mathcal{V}$, the node features $\boldsymbol{f}_i = \boldsymbol{f}_i'$ and the multisets of edge features $\{\!\{e_{ij} \mid j \in \mathcal{N}_i\}\!\} = \{\!\{e_{ij}' \mid j \in \mathcal{N}_i'\}\!\}$ are also the same. That is, $\|\boldsymbol{x}_i - \overline{\boldsymbol{x}}\| = \|\boldsymbol{x}_i' - \overline{\boldsymbol{x}'}\|$ and $\{\!\{(\|\boldsymbol{x}_i - \boldsymbol{x}_j\|, \tau_{ij}) \mid j \in \mathcal{N}_i\}\!\} = \{\!\{(\|\boldsymbol{x}_i' - \boldsymbol{x}_j'\|, \tau_{ij}') \mid j \in \mathcal{N}_i'\}\!\}$. Notice that the number of edges $|\mathcal{E}|$ is equal to $\frac{1}{2}\sum_i d_i$, where $d_i$ denotes the degree of the node $i$. Since the degrees of corresponding nodes on graphs $\mathcal{G}, \mathcal{G}'$ are equal, both graphs have the same sum of node degrees and hence the same number of edges, i.e., $|\mathcal{E}| = |\mathcal{E}'|$. Due to the generic nature of point clouds, for any edge $(i, j) \in \mathcal{E}$, the distance $\|\boldsymbol{x}_i' - \boldsymbol{x}_j'\|$ will appear exactly once in the multisets corresponding to nodes $i$ and $j$ on both graphs. This means that $(i, j)$ is also an edge in $\mathcal{E}'$ and hence the underlying graph $(\mathcal{V}, \mathcal{E})$ and $(\mathcal{V}', \mathcal{E}')$ are isomorphic. It also follows that the edge attributes of corresponding edges are the same, i.e., $\|\boldsymbol{x}_i - \boldsymbol{x}_j\| = \|\boldsymbol{x}_i' - \boldsymbol{x}_j'\|$ and $\tau_{ij} = \tau_{ij}'$. Next, we observe that the triple $(\|\boldsymbol{x}_i - \boldsymbol{x}_j\|, \|\boldsymbol{x}_i - \overline{\boldsymbol{x}}\|, \|\boldsymbol{x}_j - \overline{\boldsymbol{x}}\|)$ uniquely determines the distance between projected points $p_{\overline{\boldsymbol{x}}}(\boldsymbol{x}_i), p_{\overline{\boldsymbol{x}}}(\boldsymbol{x}_j)$ on the unit sphere as we show in Lemma C.1 below. Therefore, the equality between triples $(\|\boldsymbol{x}_i - \boldsymbol{x}_j\|, \|\boldsymbol{x}_i - \overline{\boldsymbol{x}}\|, \|\boldsymbol{x}_j - \overline{\boldsymbol{x}}\|) = (\|\boldsymbol{x}_i' - \boldsymbol{x}_j'\|, \|\boldsymbol{x}_i' - \overline{\boldsymbol{x}'}\|, \|\boldsymbol{x}_j' - \overline{\boldsymbol{x}'}\|)$ implies $\|p_{\overline{\boldsymbol{x}}}(\boldsymbol{x}_i) - p_{\overline{\boldsymbol{x}}}(\boldsymbol{x}_j)\| = \|p_{\overline{\boldsymbol{x}'}}(\boldsymbol{x}_i') - p_{\overline{\boldsymbol{x}'}}(\boldsymbol{x}_j')\|$. Combined with the identity $\tau_{ij} = \tau_{ij}'$ and the fact that $(\mathcal{V}, \mathcal{E})$ and $(\mathcal{V}', \mathcal{E}')$ are isomorphic, we can conclude that $(\mathcal{V}, \mathcal{E}, p_{\overline{\boldsymbol{x}}}(\boldsymbol{X}))$ and $(\mathcal{V}', \mathcal{E}', p_{\overline{\boldsymbol{x}'}}(\boldsymbol{X}'))$ are geometrically isomorphic using Lemma 3.5 and Theorem 2.1. Finally, because we can recover the original coordinates $\boldsymbol{X}$ from the projected points $p_{\overline{\boldsymbol{x}}}(\boldsymbol{X})$ by $\|\boldsymbol{x}_i - \overline{\boldsymbol{x}}\| \cdot p_{\overline{\boldsymbol{x}}}(\boldsymbol{x}_i) = \boldsymbol{x}_i$, $(\mathcal{V}, \mathcal{E}, \boldsymbol{X}, \boldsymbol{F})$ and $(\mathcal{V}', \mathcal{E}', \boldsymbol{X}', \boldsymbol{F}')$ are also geometrically isomorphic. $\square$

**Lemma C.1.** *The triple* $(\|\boldsymbol{x}_i - \boldsymbol{x}_j\|, \|\boldsymbol{x}_i - \overline{\boldsymbol{x}}\|, \|\boldsymbol{x}_j - \overline{\boldsymbol{x}}\|)$ *uniquely determines the distance between projected points* $p_{\overline{\boldsymbol{x}}}(\boldsymbol{x}_i), p_{\overline{\boldsymbol{x}}}(\boldsymbol{x}_j)$ *on the unit sphere.*

*Proof of Lemma C.1.* We denote the projection $p_{\overline{\boldsymbol{x}}}$ by $p$ for simplicity. Notice that the angle formed by the vertices $\boldsymbol{x}_i, \overline{\boldsymbol{x}}, \boldsymbol{x}_j$ is the same as the angle formed by the vertices $p(\boldsymbol{x}_i), \overline{\boldsymbol{x}}, p(\boldsymbol{x}_j)$, since projection $p$ preserves the angle subtended by these points. Denote this angle by $\alpha$. Following the Law of Cosines, we have:

$$\cos(\alpha) = \frac{\|\boldsymbol{x}_i - \overline{\boldsymbol{x}}\|^2 + \|\boldsymbol{x}_j - \overline{\boldsymbol{x}}\|^2 - \|\boldsymbol{x}_i - \boldsymbol{x}_j\|^2}{2\|\boldsymbol{x}_i - \overline{\boldsymbol{x}}\| \cdot \|\boldsymbol{x}_j - \overline{\boldsymbol{x}}\|} \tag{6}$$

and

$$\|p(\boldsymbol{x}_i) - p(\boldsymbol{x}_j)\|^2 = \|p(\boldsymbol{x}_i) - \overline{\boldsymbol{x}}\|^2 + \|p(\boldsymbol{x}_j) - \overline{\boldsymbol{x}}\|^2 - 2\|p(\boldsymbol{x}_i) - \overline{\boldsymbol{x}}\|\|p(\boldsymbol{x}_j) - \overline{\boldsymbol{x}}\| \cos(\alpha)$$
$$= 2 - 2\cos(\alpha), \tag{7}$$

where we have used the fact that $\|p(\boldsymbol{x}_i) - \overline{\boldsymbol{x}}\| = 1$ and $\|p(\boldsymbol{x}_j) - \overline{\boldsymbol{x}}\| = 1$ as both points lie on the unit sphere. From equation (6), we can compute $\cos(\alpha)$ using the given triple ($\|\boldsymbol{x}_i - \boldsymbol{x}_j\|, \|\boldsymbol{x}_i - \overline{\boldsymbol{x}}\|, \|\boldsymbol{x}_j - \overline{\boldsymbol{x}}\|$). Substituting this value of $\cos(\alpha)$ into equation (7), we can then solve for $\|p(\boldsymbol{x}_i) - p(\boldsymbol{x}_j)\|$, which gives the distance between the projected points. $\square$

## D  MAINTAINING THE CONNECTIVITY OF SCHULL

In the SCHull graph construction presented in the main text, we assume the following condition holds:

$$\boldsymbol{x}_i \neq \overline{\boldsymbol{x}} \text{ and } p_{\overline{\boldsymbol{x}}}(\boldsymbol{x}_i) \neq p_{\overline{\boldsymbol{x}}}(\boldsymbol{x}_j) \text{ for any } i \neq j \in \mathcal{V}. \tag{8}$$

Here, we outline a strategy to maintain the connectivity and sparsity of our graph when the generic condition equation (3) does not hold.

**When $\boldsymbol{x}_i = \overline{\boldsymbol{x}}$ for some node $i$:** To preserve graph connectivity, we employ a strategy similar to $k$-nearest neighbors. We connect $\boldsymbol{x}_i$ to its nearest neighbor(s) in the point cloud. Unless there is a significant number of points exactly at the center of the point cloud, this modification will not substantially compromise sparsity.

**When $p_{\overline{\boldsymbol{x}}}(\boldsymbol{x}_i) \neq p_{\overline{\boldsymbol{x}}}(\boldsymbol{x}_j)$ for some nodes $i, j$:** In these cases, we compare the norms $\|\boldsymbol{x}_i\|$ and $\|\boldsymbol{x}_j\|$. If $\|\boldsymbol{x}_i\| > \|\boldsymbol{x}_j\|$, we treat $\boldsymbol{x}_i$ as the representative of the point $p_{\overline{\boldsymbol{x}}}(\boldsymbol{x}_i)$ on the unit sphere and then connect edges to $\boldsymbol{x}_i$ following the original SCHull graph construction. Subsequently, we connect $\boldsymbol{x}_j$ to $\boldsymbol{x}_i$. This approach ensures that sparsity is preserved.

## E  SCHULL GRAPHS OF 2D POINT CLOUDS.

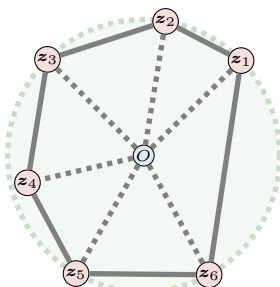

Figure 10: A visualization illustrating how the edge lengths of a polygon on the unit circle uniquely determine the spatial arrangement of its vertices.

When a point cloud lies on the plane, its associated SCHull graph corresponds to a polygon, with each vertex positioned on the unit circle, as illustrated in Fig. 10. In this case, the dihedral angles, as defined in equation (5), become zero since there is only a single facet.

Nevertheless, the SCHull graph with edge attributes $\|\boldsymbol{x}_i - \boldsymbol{x}_j\|$ and node attributes $\|\boldsymbol{x}_i - \overline{\boldsymbol{x}}\|$ provides sufficient information for a maximally expressive GNN to distinguish non-isomorphic generic point clouds on the plane. Here the genericity of point clouds (Definition 3.3) is characterized by 2D coordinates on the plane. The formal result is stated in the following theorem:

**Theorem E.1.** *Let $F$ be a maximally expressive GNN with depth $T = 1$. Then, $F$ can distinguish between the attributed SCHull graphs defined in equation (5) of any two non-isomorphic generic point clouds on the plane.*

The proof of this theorem follows a similar approach to Theorem 3.6, relying on an analogous result to Theorem 2.1 for polygons on the unit circle. Specifically, we need to show that the edge lengths of a polygon uniquely determine its structure on the unit circle, up to isometry:

**Lemma E.2.** *The coordinates of the vertices of an $n$-polygon inscribed in a circle are uniquely determined by the sequence of its edge lengths, up to isometry.*

*Proof of Lemma E.2.* Let the coordinates of the vertices on the circle be denoted as $z_1, z_2, \ldots, z_n$, ordered counterclockwise. We aim to show that the edge lengths $\|z_{i+1} - z_i\|$ for $1 \le i \le n - 1$ and $\|z_1 - z_n\|$ are sufficient to uniquely determine the vertices $z_1, z_2, \ldots, z_n$, up to isometry. Each edge length $\|z_{i+1} - z_i\|$ determines the triangle formed by $z_i, z_{i+1}$, and the circle's center $O$ using the side-side-side (SSS) triangle congruence criterion. Similarly, the edge length $\|z_1 - z_n\|$ determines the triangle formed by $z_n, z_1$, and $O$. By sequentially composing these triangles in order, the shape of the entire polygon is fully determined by Cauchy's Arm Lemma (Chern, 1967) up to isometry. A visualization is demonstrated in Fig. 10. $\square$

*Proof of Theorem E.1.* Following the same steps as in the proof of Theorem 3.6, we first establish that the underlying graphs $(\mathcal{V}, \mathcal{E})$ and $(\mathcal{V}', \mathcal{E}')$. Additionally, the distances between corresponding nodes satisfy $\|p_{\overline{x}}(x_i) - p_{\overline{x}}(x_j)\| = \|p_{\overline{x'}}(x_i') - p_{\overline{x'}}(x_j')\|$. From the graph structure of the polygons, it follows that the sequence of edge lengths is identical for the two graphs. By Lemma E.2, this implies that the projected point clouds on the circle are identical up to isometry. Using the node features $\{\|x_i - \overline{x}\|\}$ and $\{\|x_i' - \overline{x'}\|\}$, we conclude that the underlying point clouds are identical up to isometry. $\square$

## F    RIGIDITY OF ATTRIBUTED SCHULL GRAPHS WITHOUT NODE ATTRIBUTES

In Section 3.3, we demonstrated that the attributed design defined in equation (5) is sufficient for maximally expressive GNNs to distinguish non-isomorphic generic point clouds in Theorem 3.6. Here, we investigate when the node attributes $f_i, \|x_i - \overline{x}\|$, representing the original distance of each point to the center of the point cloud, can be omitted. We will show that edge lengths and dihedral angles are sufficient to guarantee the rigidity of SCHull graphs of generic point clouds under the following additional generic assumption:

$$\text{Any four projected points } \{p_{\overline{x}}(x_i)\} \text{ are not coplanar.} \tag{9}$$

Point clouds that do not satisfy this condition are included in a finite union of zero sets of analytic functions, which has Lebesgue measure zero (Mityagin, 2015).

Specifically, we aim to prove the following theorem:

**Theorem F.1.** *Let $F$ be a maximally expressive GNN with depth $T = 1$. Then $F$ can distinguish between the SCHull graphs using only edge-attributes defined in equation (5) of any two non-isomorphic generic point clouds satisfying equation (9).*

Note that the proof of Theorem 3.6 relies on the rigidity of strictly-convex geometric polyhedral graphs stated in Theorem 2.1. Similarly, our proof of Theorem F.1 relies on the following rigidity theorem for geometric polyhedral graphs from (Cho & Kim, 2023):

**Theorem F.2.** *If a geometric, possibly nonconvex and self-intersecting, polyhedral graph satisfies that (i) every face is convex, (ii) there are no coplanar adjacent faces and no set of seven vertices lies on the same plane, and (iii) any triple of vertices is not collinear, then its geometric arrangement is uniquely determined up to rigid motions by its dihedral angles and edge lengths.*

To prove Theorem F.1, it suffices to show that generic point clouds satisfying equation (9) fulfill the required conditions in Theorem F.2.

First, the genericity of point clouds ensures the conditions (ii) and (iii), as they can be expressed as polynomials of points' coordinates with rational coefficients. To demonstrate that all faces on the

SCHull graph are convex, we observe that since any four projected points $\{p_{\overline{\boldsymbol{x}}}(\boldsymbol{x}_i)\}$ are not coplanar, their convex hull can only have triangles as its faces. This implies that the faces of the SCHull graph are all triangular, and triangles are inherently convex.

# G   Algorihtms and Architecture Details

## G.1   Graph Construction Algorithms

**Radial cutoff.**   The Radial cutoff graph construction is implemented using the PyTorchGeometric (Fey & Lenssen, 2019) data transform `RadiusGraph`. This constructs the set of edges $\{e_{ij} \mid ||x_i - x_j|| < r\}$ for a fixed radius $r$. In the NestedSquares dataset $r = 4$, this is the side length of the outer square. In all other tasks, where the baseline models use RadiusGraph to construct the graph, we use the default radius $r$.

$k$**NN.**   The $k$NN graph construction is implemented using the PyTorchGeometric (Fey & Lenssen, 2019) data transform `KNNGraph`. This constructs a directed set of edges such that each node has exactly $k$ neighbors. The graph is then made undirected. In all tasks $k = 3$.

**Voronoi.**   The Voronoi graph construction is implemented using the SciPy (Virtanen et al., 2020) spatial transform `Voronoi`. This constructs an undirected graph from the ridges of the Voronoi diagram.

## G.2   Message Passing Neural Network

The following message-passing neural network:

$$\boldsymbol{m}_{ij} = \phi_m\big(\boldsymbol{h}_i^l, \boldsymbol{h}_j^l, \boldsymbol{e}_{ij}\big)$$
$$\boldsymbol{m}_i = \sum_{j \in \mathcal{N}(i)} \boldsymbol{m}_{ij}$$
$$\boldsymbol{h}_i^{l+1} = \phi_h(\boldsymbol{h}_i^l, \boldsymbol{m}_i)$$

is used for the experiments in Section 3.3. In particular, for the architecture MPNN+SCHull, we utilize the attributed version of SCHull defined in equation (5). For the base MPNN architecture, the edge attributes are discarded, i.e., $\boldsymbol{m}_{ij} = \phi_m\big(\boldsymbol{h}_i^l, \boldsymbol{h}_j^l\big)$.

# H   Benchmark Details

## H.1   NestedSquares

The NestedSquares dataset is a graph property prediction task designed to test the expressivity of geometric GNNs. The training, validation, and test splits contain rotated graphs depicted in Fig. 4. There are 6 training graphs, 2 test graphs, and 2 validation graphs, and they contain randomly rotated graphs. The training procedure uses the Adam optimizer to minimize the mean squared error (MSE) loss between the model predictions and ground truth graph labels. The training uses the Adam optimizer with a learning rate of $1e$-4 and a learning rate scheduler, which reduces by a factor of $0.9$, has a patience of $25$, and a minimum lr of $1e$-5. The model is trained for $100$ epochs, with the test loss reported for the epoch with the best validation loss.

## H.2   MD17

The MD17 dataset is a node property prediction task, which contains a variety of molecular dynamics trajectories. Both the training and validation sets contain 1K samples, with the remaining data used for testing. The training procedure uses the Adam optimizer to minimize the L1 loss between the model predictions and ground truth molecular energy and forces. The training uses the Adam optimizer with a learning rate of $5e$-4 and a learning rate scheduler which reduces by a factor of $0.5$ and has a patience of $50$. The model is trained for $1100$ epochs, with the test loss reported for the final epoch.

### H.3 Protein Dataset

**Fold dataset**. We utilize the same dataset as that used by Hou et al. (2018); Wang et al. (2022a). The dataset includes 16,292 proteins across 1,195 folds. To assess generalization performance, three test sets are employed: Fold, where proteins from the same superfamily are excluded from training; Superfamily, where proteins from the same family are not included in training; and Family, where proteins from the same family are part of the training data. Among these, Fold poses the greatest challenge, as its proteins differ the most from those in the training set. For this task, 12,312 proteins are used for training, 736 for validation, 718 for Fold, 1,254 for Superfamily, and 1,272 for Family.

**Reaction dataset**. For the reaction classification task, 3D structures of 37,428 proteins corresponding to 384 EC numbers are obtained from PDB (Berman et al., 2000), with EC annotations for each protein retrieved from the SIFTS database (Dana et al., 2019). The dataset is divided into 29,215 proteins for training, 2,562 for validation, and 5,651 for testing. Each EC number is represented across all three splits, and protein chains sharing more than 50% sequence similarity are grouped.

**LBA dataset**. Following (Jing et al., 2020), we perform ligand binding affinity predictions on a subset of the commonly-used PDBbind refined set (Wang et al., 2004; Liu et al., 2015). The curated dataset of 3,507 complexes is split into train/val/test splits based on a 30% sequence identity threshold to verify the model generalization ability for unseen proteins. For a protein-ligand complex, we predict the negative log-transformed binding affinity $pK = -\log_{10}(K)$ in Molar units.

## I  Hyperparameter Details

### I.1  NestedSquares

Table 9 reports the hyperparameters for each invariant GNN which are adjusted from the baseline models. Additionally, we find that models are particularly sensitive to the cutoff which serves as a soft threshold parameter in each of the models. We report the results from the best cutoff using a grid search from 1 to 10. The chosen cutoff parameter is reported in Table 10.

|  | hidden_features | num_radial | num_spherical |
|---|---|---|---|
| DimeNet++ | 128 | 6 | 7 |
| SphereNet | 256 | 6 | 7 |
| ComENet | 128 | 8 | 8 |
| MPNN+SCHull | 256 | 32 | 3 |

Table 9: Hyperparameter selection for the NestedSquares dataset.

|  | Radial cutoff | $k$NN ($k=3$) | Voronoi | SCHull (ours) |
|---|---|---|---|---|
| MPNN | 8 | 6 | 7 | 8 |
| DimeNet++ | 3 | 3 | 3 | — |
| SphereNet | 9 | 9 | 9 | — |
| ComENet | 8 | 8 | 7 | — |

Table 10: Cutoff hyperparameter in the NestedSquarse task for each model and graph connectivity.

### I.2  MD17

Model and training hyperparameters for the MD17 task are listed in Table 11.

### I.3  Protein Dataset

Model and training hyperparameters search space for the protein tasks are listed in Table 12

| Hyperparameter | Values/Search Space |
|---|---|
| Number of layers | 4, 6 |
| Hidden channels | 256 |
| Number of radial basis | 32, 64 |
| Cutoff | 6, 8, 10 |
| Epochs | 1200 |
| Batch size | 64 |
| Learning rate | 5e-4 |
| Learning rate scheduler | steplr |
| Learning rate decay factor | 0.5 |
| Learning rate decay epochs | 180 |
| Weight decay rate | 1e-12 |

Table 11: Model and training hyperparameters search space for MD17 tasks.

| Hyperparameter | Values/Search Space | | |
|---|---|---|---|
| | React | Fold | LBA |
| Number of layers | 3, 4, 5 | 3, 4, 5 | 3, 4, 5 |
| Hidden channels | 64, 128, 256 | 64, 128, 256 | 128, 192, 256 |
| Cutoff | 6, 8, 10 | 6, 8, 10 | 6, 8, 10 |
| SCHull+Cutoff | 6, 7, 8 | 6, 7, 8 | 6, 8 |
| Dropout | 0.2, 0.3, 0.5 | 0.2, 0.3, 0.5 | 0.2, 0.3 |
| Epochs | 500, 1000 | 500, 1000 | 300, 500 |
| Batch size | 16, 32 | 16, 32 | 8, 16, 32 |
| Learning rate | 1e-4, 5e-4 | 1e-4, 5e-4 | 5e-5, 1e-4, 2e-4 |
| Learning rate scheduler | steplr | steplr | steplr |
| Learning rate decay factor | 0.5 | 0.5 | 0.5 |
| Learning rate decay epochs | 50, 100 | 100, 200 | 50, 100 |

Table 12: Model and training hyperparameters for protein-related datasets.

