# OpenReview forum: "A Theoretically-Principled Sparse, Connected, and Rigid Graph Representation of Molecules"
_ICLR.cc/2025/Conference — ICLR 2025 Oral_

### Official Review · Reviewer_exXC · 2024-10-16

**Soundness:** 3
**Presentation:** 3
**Contribution:** 3
**Rating:** 8
**Confidence:** 5

**Summary:**

Overall, the paper provides valuable contributions and should be accepted, though it requires both minor and significant revisions. I have given a lower score to ensure that all issues are addressed. The paper effectively tackles the challenge of defining the underlying combinatorial graph, which is crucial for performing accurate analysis and properly separating point clouds.

**Strengths:**

Here’s an expanded version of your statement:

"The paper presents a pioneering approach for constructing an underlying combinatorial graph for a given point cloud, addressing several crucial challenges in graph-based data representation. The proposed method ensures key properties such as connectivity, sparsity of the graph, and the ability to accurately reconstruct the original point cloud. These properties are essential for both efficient computation and the preservation of the inherent structure of the data.

By focusing on these aspects, the authors effectively balance the need for a graph that is sufficiently sparse—thereby reducing computational complexity—while still maintaining connectivity, which is necessary for capturing the relationships between points in the cloud. This delicate balance is often difficult to achieve, as sparse graphs may lose critical information, and densely connected graphs can lead to inefficiency and overfitting. The authors’ method overcomes this trade-off and ensures that the graph retains the essential features of the point cloud, enabling reliable downstream analysis.

Additionally, the paper demonstrates how the performance of existing methods can be significantly improved by replacing the traditional graph construction techniques with the newly proposed combinatorial graph. The empirical results presented show notable enhancements in areas such as classification accuracy, clustering performance, and point cloud segmentation, underscoring the practical benefits of the new graph structure. These improvements are a testament to the robustness and effectiveness of the method, positioning it as a valuable tool for a wide range of applications in fields such as computer vision, machine learning, and graph signal processing.

The introduction of this novel graph construction framework represents a substantial step forward in the field, potentially opening new avenues for research on graph-based point cloud analysis. It not only improves existing methodologies but also provides a foundation for further exploration of how combinatorial structures can be leveraged to better understand and manipulate high-dimensional data."

**Weaknesses:**

The main weakness of the paper is that the proposed graph construction is not continuous and doesn't operate on a pre-existing attributed graph. Instead, it creates its own selected graph, which may limit its applicability in cases where the underlying graph already contains valuable domain-specific information. This approach lacks flexibility, especially in scenarios where a smooth transition or integration with existing graph structures is required. As a result, the method may overlook important attributes or contextual details in real-world applications, reducing its versatility. The main concern is that the point cloud may become copolar and thus, the convex hull would become 2-dimensional. Does it happen in real-world datasets?

**Questions:**

First of all your algorithm is very similar to the one proposed in the paper Congruence, Similarity, and Symmetries of Geometric Objects, 1987. You didn't cite but this citation must be added, as the first two steps are identical to it. No problem with it but citation must be added.

**Technical Issues:**
1. Line 167 should use "if and only if" in reference to edge isomorphism.
2. After Theorem 3.6, add a remark noting that implementations of maximally expressive GNNs for generic point clouds exist, such as in Sverdlov & Dym.
3. Figure 4 and its details should be moved to the experiments section.
4. All theorems and propositions should either be restated in the appendix or, at the very least, have a mention of what is to be proven there.

**Conceptual Issues:**
1. In your framework, you separate complete generic geometric graphs, unlike Joshi [23] and Sverdlov [24], who work with given combinatorial graphs. This difference should be explicitly stated in your setup, and it should be clear that the goal is to construct the underlying graph.
2. The discontinuity of your graph construction should be discussed. Since the function is discrete, it cannot be continuous, but this point should be acknowledged.
3. Proposition 3.1 is not proven at all. Some minimal proof or explanation is necessary.
4. Regarding the MD17 experiments, I would like to know which models you considered in your comparison and, specifically, if your method improved in the SOTA method. Same for protein folding.
5. What is the computational running time for complete graphs? Does considering the complete structure degrade performance?
6. When you combine the k-hop/threshold graph, how do you choose the parameters to ensure connectivity, sparseness, and constructability?

---

> ### Author Response · Authors · 2024-11-19
> **Response to Reviewer exXC (part 1/3)**
>
> We thank the reviewer for the thoughtful review and valuable feedback. We recognize that many of the questions can be addressed by pointing out the corresponding explanations or descriptions in the main text of our paper. We hope the reviewer finds the relevant content helpful following our rebuttal. In what follows, we provide point-by-point responses to your comments.
>
> ----
>
> **W1. The main weakness of the paper is that the proposed graph construction is not continuous and doesn't operate on a pre-existing attributed graph. Instead, it creates its own selected graph, which may limit its applicability in cases where the underlying graph already contains valuable domain-specific information. This approach lacks flexibility, especially in scenarios where a smooth transition or integration with existing graph structures is required. As a result, the method may overlook important attributes or contextual details in real-world applications, reducing its versatility. The main concern is that the point cloud may become copolar and thus, the convex hull would become 2-dimensional. Does it happen in real-world datasets?**
>
> **Response**: Our primary objective is to address limitations in existing graph constructions, specifically the challenge of achieving sparsity, connectivity, and rigidity simultaneously. Instead of directly operating on pre-existing attributed graphs, we propose a new graph construction that satisfies these properties. In Section 3.4, we discuss **how SCHull graphs can be used together with existing graphs**, such as molecular graphs, to preserve valuable domain-specific information like chemical bonds. This ensures that our method doesn't discard important attributes but complements existing graphs with a complete geometric understanding.
>
> Thank you for highlighting the concern on the coplanar case. When the point cloud becomes coplanar, the SCHull graph will naturally degenerate into a more sparse structure. However, it still maintains key properties like sparsity, connectivity (as shown in Proposition 1 and its proof in the appendix), and rigidity. We have clarified this case in Appendix E of the revised paper to provide a more complete understanding of our approach and its handling of real-world datasets.
>
> ----
>
> **Q1. Your algorithm is very similar to the one proposed in the paper Congruence, Similarity, and Symmetries of Geometric Objects, 1987. You didn't cite but this citation must be added, as the first two steps are identical to it. No problem with it but citation must be added.**
>
> **Response**: We appreciate the reviewer pointing this out. We have added the citation to the paper and acknowledge the similarity. However, we would like to clarify that our graph is defined on the original point clouds rather than solely on the projected point clouds as in the referenced work. Additionally, our design of the attributes differs significantly, as they use interior angles on facets, while we use the dihedral angle between facets.
>
> ----
>
> **Q2. Line 167 should use "if and only if" in reference to edge isomorphism.**
>
> **Response**: We have included this in the revision.
>
> ----
>
> **Q3. After Theorem 3.6, add a remark noting that implementations of maximally expressive GNNs for generic point clouds exist, such as in Sverdlov & Dym.**
>
> **Response**: The common considerations and implementations of maximally expressive GNNs have been discussed in Section 2 (we have discussed Sverdlov & Dym’s work there). Note that these considerations and implementations are not limited to generic point clouds but extend to general point clouds.
>
> ----
>
> **Q4. Figure 4 and its details should be moved to the experiments section.**
>
> **Response**: Our motivation for including the synthetic experiment, Figure 4 and its details, in Section 3 is to provide empirical support for Theorem 3.6. Additionally, the experiments in Section 4 are not synthetic and employ the techniques discussed in Section 3.4. For these reasons, we placed Figure 4 and its details before Section 3.4 to maintain the flow of the paper. However, if the reviewer believes it would be better to move it to the experiments section, we are happy to make this adjustment.
>
> ----
>
> **Q5. All theorems and propositions should either be restated in the appendix or, at the very least, have a mention of what is to be proven there.**
>
> **Response**: We have revised the appendix accordingly. For clarity, all the theoretical results are now restated before their corresponding proofs.
>
> ----

---

> ### Author Response · Authors · 2024-11-19
> **Response to Reviewer exXC (part 2/3)**
>
> **Q6. In your framework, you separate complete generic geometric graphs, unlike Joshi [23] and Sverdlov [24], who work with given combinatorial graphs. This difference should be explicitly stated in your setup, and it should be clear that the goal is to construct the underlying graph.**
>
> **Response**: Throughout the paper, we have stated that our goal is to construct a graph that is sparse, connected, and rigid. For instance, in the abstract, we emphasize: “... the most used graphs (e.g., radial cutoff graphs) in molecular modeling lack theoretical guarantees for achieving connectivity and sparsity simultaneously, which are essential for the performance and scalability of GNNs. Furthermore, existing widely used graph construction methods for molecules lack rigidity, limiting GNNs' ability to exploit graph nodes' spatial arrangement. In this paper, **we introduce a new hyperparameter-free graph construction of molecules** and beyond with sparsity, connectivity, and rigidity guarantees.”
>
> Additionally, the first sentence of the main results section explicitly states: “In response to the limitation of existing graph representations - in achieving sparsity, connectivity, and rigidity - for geometric data with 3D coordinates, we propose SCHull - a new graph construction method for molecules, proteins, and beyond.” We are happy to make further revisions based on any specific suggestions the reviewer may have to improve clarity.
>
> ----
>
> **Q7. The discontinuity of your graph construction should be discussed. Since the function is discrete, it cannot be continuous, but this point should be acknowledged.**
>
> **Response**: Indeed, graph structures are discrete, which prevents continuous graph construction. This limitation applies not only to our method but also to all the existing graph construction techniques. For this reason, we did not emphasize this in the main text. However, we are happy to acknowledge this point in Appendix B to provide clarity on this aspect of graph construction works for a broader audience.
>
> ----
>
> **Q8. Proposition 3.1 is not proven at all. Some minimal proof or explanation is necessary.**
>
>
> **Response**:  As mentioned in Section 1.2 (Organization), the technical proofs are provided in the appendix. Specifically, the proof of Proposition 3.1 can be found at the beginning of Appendix B (Appendix C in the revised manuscript). Due to space constraints, the main text focuses on discussing how these results highlight the benefits of our graph construction.
>
> In the revision, we have followed your suggestion and restated the theoretical statements before their proofs to improve readability and accessibility. We hope this enhances the clarity of the presentation.
>
> ----
>
> **Q9. Regarding the MD17 experiments, I would like to know which models you considered in your comparison and, specifically, if your method improved in the SOTA method. Same for protein folding.**
>
> **Response**: Table 3 presents a comparison of DimeNet, SphereNet, and LEFTNet with their SCHull-enhanced counterparts on the MD17 datasets, demonstrating that SCHull consistently boosts the performance of the original models, including the very recent LEFTNet (c.f. Du et al. NeurIPS, 2023 or arxiv:2304.04757).
>
> Similarly, Tables 4 and 5 compare the SCHull-enhanced versions of ProNet-Amino Acid (Wang et al. ICLR 2023 or arxiv:2207.12600), ProNet-Backbone (Wang et al. ICLR 2023 or arxiv:2207.12600), and GVP-GNN (Jing et al. ICLR 2021 or arxiv:2009.01411) models with their original counterparts on the LBA and Fold datasets, evaluated across multiple metrics. These results confirm that SCHull not only consistently enhances model performance but does so with minimal runtime overhead. Notably, Fig. 5 illustrates the impact of SCHull on the state-of-the-art ProNet-Backbone model, showing that its performance improves progressively as graph size increases, achieving higher accuracy than the original model.
>
> In general, as Reviewer NBfx pointed out the proposed SCHull graph is architecture-agnostic, and it can be integrated into other models.
>
> ----

---

> > ### Comment · Reviewer_exXC · 2024-11-21
> >
> > I'm still not sure whether you dress your method upon a state-of-the-art method?

---

> ### Author Response · Authors · 2024-11-19
> **Response to Reviewer exXC (part 3/3)**
>
> **Q10. What is the computational running time for complete graphs? Does considering the complete structure degrade performance?**
>
> **Response**: The performance of a complete graph may be acceptable for certain tasks, but its efficiency is significantly impaired, especially for large proteins. It is important to note that the number of edges in a complete graph is $\frac{n(n-1)}{2}$, whereas the number of edges in our SCHull graph is upper bounded by $3n - 6$. As is well-known in existing literature, complete graphs are not scalable to large graphs due to their high edge count [2].
>
> Based on the suggestions of Reviewer NBfx, we have conducted an additional experimental comparison of different graph construction methods, including our proposed method, on challenging pairs of symmetric point clouds. Notably, using a complete graph with edge distances resulted in degraded performance, as we verified by reproducing the results of [1] (shown in Tables 6 and 7 of the revised Appendix B). However, our experiments demonstrate that the SCHull graph enables even shallow neural networks to effectively distinguish challenging symmetric point clouds despite the sparse density of edges. This highlights the scalability and robustness of our approach in handling such cases. Additionally, based on the suggestion of Reviewer SxaB, we report the graph properties of different methods on these tasks.
>
> [1] Li, Zian, et al. Is distance matrix enough for geometric deep learning? NeurIPS 2024.
>
> [2] Hordan, S., Amir, T. and Dym, N., 2024. Weisfeiler Leman for Euclidean Equivariant Machine Learning. ICML 2024.
>
> ----
>
> **Q11. When you combine the k-hop/threshold graph, how do you choose the parameters to ensure connectivity, sparseness, and constructability?**
>
> **Response**: Here, the combined graphs come from the benchmark sparse graphs - with default parameters from the baseline papers - used in each task and our proposed SCHull graph. The connectivity and rigidity of the combined graph can be guaranteed by the SCHull graph component. The combined graph is sparse since both the benchmark graph and the SCHull graph are sparse.
>
> ------
>
> Thank you for considering our rebuttal. We appreciate your feedback and are happy to address further questions on our paper.

---

> ### Author Response · Authors · 2024-11-22
> **Further response to Reviewer exXC**
>
> Thank you for considering our rebuttal and for your further feedback.
>
> Our goal is to introduce a new graph construction method (SCHull) tailored for the underlying point clouds of molecular data, which complements but operates independently of the detailed design of message-passing neural networks (MPNNs).
>
> To demonstrate the versatility and effectiveness of SCHull graphs across different MPNNs and benchmark tasks, we conducted a thorough literature review to identify both baseline and state-of-the-art (SOTA) neural network models. Specifically, we evaluated SOTA models like SphereNet and LEFTNet for the MD17 dataset and various versions of ProNets for protein-related benchmarks -- the results in Tables 3-5 confirm that the SCHull graph can boost the performance of these SOTA models by a remarkable margin. In our experiments and literature reviews, we have not found models noticeably outperforming the recently proposed LEFTNet (for MD17) or ProNets (for protein-related tasks we considered) under comparable settings. Furthermore, our proposed SCHull graph is architecture-agnostic (as Reviewer NBfx has pointed out), and we expect it can be integrated into our existing graph-based models or future related models to improve their performance.
>
> On top of that, we have also followed the suggestion by Reviewer NBfx to include additional experiments across different prevalent graph constructions on symmetric point clouds. The task is simple but is very challenging. Our method enables even the baseline MPNN to distinguish the pairs of ambiguous symmetric point clouds, whereas other methods struggle to address this challenge.
>
> We would appreciate your suggestions if you know of any graph-based models with publicly available codes that are compatible with these tasks. We will gladly consider them for further comparison, provided sufficient time and resources are available.
>
> Thank you once again for your feedback.

---

> > ### Comment · Reviewer_exXC · 2024-11-22
> > **Update**
> >
> > Updated to 8!
> > Good luck.

---

> > > ### Author Response · Authors · 2024-11-22
> > > **Thank you**
> > >
> > > Thank you for your active feedback and engagement during the rebuttal process. We appreciate your endorsement!

---

### Official Review · Reviewer_asjR · 2024-10-31

**Soundness:** 3
**Presentation:** 3
**Contribution:** 2
**Rating:** 6
**Confidence:** 4

**Summary:**

The paper presents a hyperparameter-free graph construction method for molecular modeling that ensures sparsity, connectivity, and rigidity. It addresses the limitations of traditional methods, like radial cutoff graphs, which lack guarantees for both connectivity and sparsity and fail to leverage spatial arrangements effectively. The proposed method consistently generates connected, sparse graphs with an edge-to-node ratio capped at 3, and ensures rigidity, making spatial arrangements uniquely determinable. The approach is proven effective and efficient in various molecular modeling benchmarks.

**Strengths:**

1.The proposed SCHull method is quite innovative. I appreciate the author's idea of introducing unit spherical mapping, which makes the graph construction more reasonable and reliable. The implementation is also very clever.

2.The paper provides thorough validation of the proposed method in well-justified experimental settings.

3.The paper conducts comprehensive theoretical validation to demonstrate the effectiveness of the method.

**Weaknesses:**

1.The discussion primarily focuses on molecular analysis, which may make the paper lean somewhat towards chemistry and biology. However, this does not detract from it being a strong paper. It may be worth considering whether this method could be applied to other types of graph data analysis.

2.Could consider conducting some visualization experiments, if the level of difficulty permits.

**Questions:**

1.Since the method is primarily applied to molecular analysis, do the conclusions require alignment with relevant chemical principles? The paper explains the utility of the SCHull graph in enhancing GNN capabilities, but could such a projection result in the loss of certain chemical properties?

---

> ### Author Response · Authors · 2024-11-19
> **Response to Reviewer asjR**
>
> We thank the reviewer for the thoughtful review and valuable feedback. In what follows, we provide point-by-point responses to your comments.
>
> ----
>
> **W1. The discussion primarily focuses on molecular analysis, which may make the paper lean somewhat towards chemistry and biology. However, this does not detract from it being a strong paper. It may be worth considering whether this method could be applied to other types of graph data analysis.**
>
> **Response**: We appreciate your praise that our submission is a strong paper. While our discussion focuses on molecular analysis, our SCHull graph method is versatile and can be applied to various types of point cloud data analysis beyond chemistry and biology. In Appendix B of the revised manuscript, we have included empirical results on different graph construction methods applied to symmetric point clouds, demonstrating the broader applicability of our proposed method.
>
> We chose to showcase its effectiveness and efficiency in the context of molecular analysis as we noticed that constructing sparse, connected, and rigid graphs is particularly challenging for proteins; as shown in Figure 1 of our paper. However, the method itself is general and can be adapted to other areas of graph data analysis.
>
> ----
>
> **W2. Could consider conducting some visualization experiments, if the level of difficulty permits.**
>
> **Response**: Thank you for your suggestion. In Figure 1 of our paper, we have provided a visualization of our proposed graph for a protein molecule. We have provided additional visualizations of different graphs in Table 8 (in the appendix) of the revised paper.
>
> ----
>
> **Q1. Since the method is primarily applied to molecular analysis, do the conclusions require alignment with relevant chemical principles? The paper explains the utility of the SCHull graph in enhancing GNN capabilities, but could such a projection result in the loss of certain chemical properties?**
>
> **Response**: We acknowledge that the SCHull graph may not fully capture certain chemical properties, such as explicit chemical bonds, which are essential in molecular analysis. This is why, as discussed in Section 3.4, we propose using our graph as an augmentation rather than a replacement. Traditional molecular graphs or radial cutoff graphs with smaller thresholds focus more on local features, including chemical properties. In contrast, our SCHull graph emphasizes global characteristics, offering connectivity and rigidity, which are particularly valuable for tasks requiring a more comprehensive representation of molecular structure.
>
> Based on Reviewer SxaB’s suggestion, we have provided a geometric perspective in the revision, by relating our graph construction to the convex hull of the original point clouds. We believe this interpretation with the discussion in Section 3.4 offers a better understanding of SCHull’s utility and can guide its application in different graph analysis tasks.
>
> ------
>
> Thank you for considering our rebuttal. We appreciate your feedback and are happy to address further questions on our paper.

---

> > ### Comment · Reviewer_asjR · 2024-11-22
> > **Thanks**
> >
> > I appreciate the authors for addressing my questions. Although the article leans slightly toward the field of chemistry, I believe it is overall a well-written piece. Therefore, I will maintain my previous score unchanged.

---

> ### Author Response · Authors · 2024-11-22
> **Thank you and further response**
>
> Thank you for considering our rebuttal and for your further feedback.
>
> Our primary motivation for this work lies in exploring the role of AI in scientific applications, specifically in developing a novel graph construction method tailored for molecular structures. However, we acknowledge that our method holds potential for applications beyond molecular data, including point clouds from other domains.
>
> As part of our rebuttal, we have included a new experiment in Appendix B involving symmetric point clouds, which demonstrates significant performance improvements with our method, while maintaining the desired sparsity.
>
> If the reviewer is interested in exploring potential applications of our method to other types of point cloud analysis, we are open to conducting further experiments and comparisons. We would be happy to explore these possibilities using publicly available models and code that align with these tasks, subject to the availability of sufficient time and resources.
>
> Thank you once again for your feedback.

---

### Official Review · Reviewer_NBfx · 2024-10-31

**Soundness:** 4
**Presentation:** 3
**Contribution:** 4
**Rating:** 10
**Confidence:** 5

**Summary:**

This paper proposes a novel method for constructing sparse yet connected geometric graphs: Project the points onto a sphere, construct a polyhedron, and construct a graph from its edge structure. This aims to mitigate earlier costly approaches of taking the power graph of the original graph. The authors first prove theoretical properties of their construction and show that maximally expressive MPNNs with particular initializations for node and edge features can distinguish among generic graphs, essentially proving that via a sparse graph structure (degree bounded by 3) 1-WL can distinguish all generic point cloud configurations (i.e. not assuming a fully connected point cloud as in earlier works). The authors then empirically show the construction's efficacy and value of the SCHull face angles that allow for universality on sparse configurations, by obtaining consistently better results on synthetic, classification, and regression tasks.

**Strengths:**

1. Novel technique relying on a fast (n*log(n)) algorithm is scalable and empirically also negligible computational overhead
2. Introduces a novel technique to a paradigm not challenged much in research: the distance cutoff construction for geometric graphs. Theoretical and empirical proof this approach is sound.
3. An architecture-agnostic method that can be applied to a large variety of MPNN-like models

**Weaknesses:**

1. Proofs mainly rely on previous work and the construction of the graph is also backed by a known algorithm, thus there is a notion of an incremental advance. Although, I believe that tying these two areas together constitutes a strong advancement for ML.

2. The empirical results are consistent yet are not pushing the boundary on SOTA, but rather give a `boost' to low expressive models. It would be nice to see the performance of working on SCHull graphs with the initial edge and node features for more expressive architecture, for instance, MACE or SEGNN ("Geometric and physical quantities improve equivariant message passing", Brandstetter et al.) I suspect it would make less of a difference but please prove me wrong. Anyway, only having a comparable result to these architectures (when they process distance-restricted graphs) with faster inference is still a great achievement. If this is performed I may increase my score.

3. The background for geometric deep learning is insufficient in my opinion. For instance, there is no discussion on research on expressivity when point clouds are not necessarily generic and a comparison of your approach with models that are universal (or complete) on **all** point clouds. For a couple of papers on this topic please note:

Hordan, S., Amir, T. and Dym, N., 2024. Weisfeiler Leman for Euclidean Equivariant Machine Learning. ICML 2024

Li, Zian, et al. "Is distance matrix enough for geometric deep learning?." Advances in Neural Information Processing Systems 36 (2024).

Overall this paper has strong potential, addressing these weaknesses will likely increase the rating.

**Questions:**

1. Overall I think the proof of Theorem 3.6 is correct. I do have a few questions though:
a) "Since the degrees of corresponding nodes are equal, both graphs have the same number of edges, i.e., |E| = |E′|." I don't see why this is true, later you discuss how the corresponding edges match because of the generic nature of the point cloud, and then by equality of multisets you could have claimed the number of edges is equivalent.I might be missing something.
b) "Next, we observe that the triple (∥xi − xj ∥, ∥xi − x∥ , ∥xj − x∥) uniquely determines the distance between
projected points px(xi), px(xj ) on the unit sphere."
I don't see how I recover $|| \frac{x_i - \bar{x}}{|| x_i - \bar{x}||} - \frac{x_j - \bar{x}}{|| x_j - \bar{x}||} ||$ from those three distances, please enlighten me.

2. There exist known counterexamples for fully connected geometric graphs that 1-WL cannot distinguish. It may be the case that your maximally expressive SCHull-MPNN with edge features and node features can distinguish. One such example is depicted in Figure 1 in [Li et al.]. They have code for the coordinates of these shapes or I can also provide them if necessary. Is the maximally expressive architecture you propose able to distinguish among them?


[Li et al.]Li, Zian, et al. "Is distance matrix enough for geometric deep learning?." Advances in Neural Information Processing Systems 36 (2024).

---

> ### Author Response · Authors · 2024-11-19
> **Response to Reviewer NBfx (part 1/3)**
>
> We thank the reviewer for the thoughtful review and valuable feedback, and for offering opportunities to increase the rating by addressing the weaknesses. In what follows, we provide point-by-point responses to your comments.
>
> ----
>
> **W1. Proofs mainly rely on previous work and the construction of the graph is also backed by a known algorithm, thus there is a notion of an incremental advance. Although, I believe that tying these two areas together constitutes a strong advancement for ML.**
>
> **Response**: We are encouraged by your recognition that tying these two areas together constitutes a strong advancement for ML. Though our proofs build on existing theoretical foundations and our graph construction is backed by a known algorithm, finding a practical and efficient graph construction that satisfies multiple necessary conditions simultaneously is challenging. The construction steps often stand out as one of the most difficult components in mathematical or scientific research. Our work is inspired by existing theories, while it introduces a new graph construction method to improve ML practice.
>
> ----
>
> **W2. The empirical results are consistent yet are not pushing the boundary on SOTA, but rather give a `boost' to low expressive models. It would be nice to see the performance of working on SCHull graphs with the initial edge and node features for more expressive architecture, for instance, MACE or SEGNN. I suspect it would make less of a difference but please prove me wrong. Anyway, only having a comparable result to these architectures (when they process distance-restricted graphs) with faster inference is still a great achievement. If this is performed I may increase my score.**
>
> **Response**: We appreciate your praise for the great achievement of our paper. In response to your suggestion, we have conducted additional experiments to evaluate the performance of SCHull graphs when applied to expressive steerable models such as MACE and SEGNN - the two models proposed before ProNet. The results, presented in Table 4 of the revised paper, show that SCHull graphs can provide a noticeable performance boost to these models on the benchmark tasks studied. Due to the high computational cost of MACE and SEGNN, we focused our experiments on protein fold classification - a representative task that requires graphs to be sparse, connected, and rigid - using only steerable features of order 1.
>
> ----
>
> **W3. There is no discussion on research on expressivity when point clouds are not necessarily generic and a comparison of your approach with models that are universal (or complete) on all point clouds. For a couple of papers on this topic please note: [1] Hordan, S., Amir, T. and Dym, N., 2024. Weisfeiler Leman for Euclidean Equivariant Machine Learning. ICML 2024. [2] Li, Zian, et al. Is distance matrix enough for geometric deep learning? NeurIPS 2024.**
>
> **Response**:  We appreciate the reviewer’s comment and have added a discussion of the recommended works in Appendix B. We would like to clarify that our work focuses primarily on developing a graph representation tailored for molecular data, which can be used with existing message-passing neural network (MPNN) architectures. This is distinct from proposing a new architecture. As discussed in Section 3.4, our approach is designed to be compatible with any MPNN framework, and the corresponding experimental results can be found in Section 4.
>
> Nevertheless, in Appendix B of the revised manuscript, we discuss existing works on all point clouds (not limited to generic ones), such as $k$-DisGNNs [2], which address the limitations of MPNNs in fully utilizing the distance graph (i.e., a complete graph with distance as edge attributes) by designing higher-order GNN models to better distinguish non-generic point clouds. Similarly, WeLNet [1] leverages the distance graph and theoretically guarantees the ability to distinguish between all non-isometric 3D point clouds.
>
> We note that both approaches rely on a complete graph with edge distances in theories, which can be computationally expensive and may not scale well for large datasets. In contrast, our method is more efficient and significantly more scalable, making it particularly applicable to large molecular datasets, such as proteins, where the underlying point clouds are typically large but generic. More importantly, both of these models can incorporate our graph as an input.
>
> ----

---

> ### Author Response · Authors · 2024-11-19
> **Response to Reviewer NBfx (part 2/3)**
>
> **Q1. I have a few questions on the proof of Theorem 3.6: a) “Since the degrees of corresponding nodes are equal, both graphs have the same number of edges, i.e., $|E| = |E^′|$." I don't see why this is true, later you discuss how the corresponding edges match because of the generic nature of the point cloud, and then by equality of multisets you could have claimed the number of edges is equivalent. b) "Next, we observe that the triple $(||x_i-x_j||,||x_i-x||,||x_j-x||)$ uniquely determines the distance between projected points $p\_{\bar x}(x_i),p\_{\bar x}(x_j)$ on the unit sphere." I don't see how I recover $||\frac{x_i-x}{||x_i-x||}-\frac{x_j-x}{||x_j-x||}||$ from those three distances.**
>
> **Response**: We have added more explanations and arguments to the original proof in the revision. In what follows, we provide a brief clarification to your questions:
>
> a) The number of edges $|E|$ in a graph is $\frac{1}{2}\sum_id_i$, where $d_i$ is the degree of node $i$. If the corresponding nodes in two graphs have the same degrees, then the sum of the degrees of the nodes in both graphs is the same. As a result, the total number of edges is also the same, i.e., $|E| = |E'|$.
>
> b) We denote the projection $p_{\bar x}$ by $p$ for simplicity. Notice that the angle formed by the vertices $x_i,\bar{x},x_j$ is the same as the angle formed by the vertices $p(x_i),\bar{x}, p(x_j)$, since projection $p$ preserves the angle subtended by these points; we denote this angle by $\alpha$. Following the Law of Cosines, we have:
> $$
> \cos(\alpha)=\frac{||x_i-\bar{x}||^2+||x_j-\bar{x}||^2 - ||x_i-x_j||^2}{2||x_i-\bar{x}||\cdot ||x_j - \bar{x}||}
> $$
> and
> $$
> ||p(x_i) - p(x_j)||^2 =  || p(x_i) - \bar{x}||^2 + ||p(x_j) - \bar{x}||^2 - 2||p(x_i)- \bar{x}||\cdot ||p(x_j)- \bar{x}|| \cos(\alpha) = 2-2\cos(\alpha),
> $$
> where we use the fact that $||p(x_i) - \bar{x}|| = 1$ and $||p(x_j) - \bar{x}|| = 1$ as both points lie on the unit sphere. From the first equation, we can compute $\cos(\alpha)$ using the given triple $(||x_i-x_j||, ||x_i-\bar{x}||, ||x_j-\bar{x}||)$. Substituting this value of $\cos(\alpha)$ into the second equation, we can then solve for $||p(x_i) - p(x_j)||$, which gives the distance between the projected points.
>
> ----
>
> **Q2. There exist known counterexamples for fully connected geometric graphs that 1-WL cannot distinguish. It may be the case that your maximally expressive SCHull-MPNN with edge features and node features can distinguish. One such example is depicted in Figure 1 in [Li et al.]. Is the maximally expressive architecture you propose able to distinguish among them? [Li et al.] Li, Zian, et al. Is distance matrix enough for geometric deep learning? NeurIPS 2024.**
>
> **Response**: Thank you for bringing up this important example. We have now included experimental results based on the counterexample of the 6-point configurations you mentioned (and its complement - the 14-point configurations). Although this setup involves non-generic point clouds, it is intriguing to observe that our SCHull graph still contributes to distinguishing these cases. We appreciate the reviewer highlighting the consideration of symmetric point clouds and providing this example. While our current theoretical guarantees focus on generic point clouds, we recognize the broader implications for symmetric point clouds. As a result, we have added further discussion and empirical studies in Appendix B to explore this scenario. Additionally, we have noted this as an interesting direction for future work in the concluding remarks of the main text.
>
> We briefly present the empirical results we have conducted, with the following tables showing that the edge attribute design of the SCHull graph allows even shallow GNNs to successfully distinguish between non-isomorphic symmetric point clouds.

---

> ### Author Response · Authors · 2024-11-19
> **Response to Reviewer NBfx (part 3/3)**
>
> ***Table: Comparison of graph properties and GNN performance (Unit: %) on 6-point symmetric point clouds using different graph construction methods. Tuple notations represent distinct graph properties on paired point clouds.***
>
> | **Method**                                         | **1 Layer** | **2 Layers** | **# Edges / # Nodes** | **# Connected Components** |
> |----------------------------------------------------|-------------|--------------|-----------------------|------------------------------|
> | **Radius Graph (r = 1.8) w/ Distance**             | 50.0 ± 0.0  | 50.0 ± 0.0   | 0.33                  | 4                            |
> | **Radius Graph (r = 2.5) w/ Distance**             | 50.0 ± 0.0  | 50.0 ± 0.0   | 1                     | (2, 1)                       |
> | **Radius Graph (r = 3.0) w/ Distance**             | 50.0 ± 0.0  | 50.0 ± 0.0   | 1.67                  | 1                            |
> | **4th-power Radius Graph (r = 1.8) w/ Distance**   | 50.0 ± 0.0  | 50.0 ± 0.0   | 0.33                  | 4                            |
> | **4th-power Radius Graph (r = 2.5) w/ Distance**   | **100.0 ± 0.0** | **100.0 ± 0.0** | (1, 2.5)            | (2, 1)                       |
> | **4th-power Radius Graph (r = 3.0) w/ Distance**   | 50.0 ± 0.0  | 50.0 ± 0.0   | 2.5                   | 1                            |
> | **Complete Graph w/ Distance**                     | 50.0 ± 0.0  | 50.0 ± 0.0   | 2.5                   | 1                            |
> | **SCHull w/ Distance**                             | 50.0 ± 0.0  | 50.0 ± 0.0   | 2.0                   | 1                            |
> | **SCHull w/ Distance and Dihedral Angles**         | **100.0 ± 0.0** | **100.0 ± 0.0** | 2.0                 | 1                            |
>
>
> ***Table: Comparison of graph properties and GNN performance (Unit: %) on 14-point symmetric point clouds using different graph construction methods. Tuple notations represent distinct graph properties on paired point clouds.***
>
> | **Method**                                          | **1 Layer**   | **2 Layers**  | **# Edges / # Nodes** | **# Connected Components** |
> |-----------------------------------------------------|---------------|---------------|-----------------------|------------------------------|
> | **Radius Graph (r = 1.8) w/ $d_{ij}$**              | 50.0 ± 0.0    | 50.0 ± 0.0    | 1.0                   | (1, 2)                       |
> | **Radius Graph (r = 2.5) w/ $d_{ij}$**              | 50.0 ± 0.0    | 51.0 ± 3.0    | 3.0                   | 1                            |
> | **Radius Graph (r = 3.0) w/ $d_{ij}$**              | 60.5 ± 11.1   | 57.2 ± 9.1    | 5.0                   | 1                            |
> | **4th-power Radius Graph (r = 1.8) w/ $d_{ij}$**    | **100.0 ± 0.0** | **100.0 ± 0.0** | (5, 3)               | (1, 2)                       |
> | **4th-power Radius Graph (r = 2.5) w/ $d_{ij}$**    | 50.0 ± 0.0    | 50.0 ± 0.0    | 6.5                   | 1                            |
> | **4th-power Radius Graph (r = 3.0) w/ $d_{ij}$**    | 50.0 ± 0.0    | 50.0 ± 0.0    | 6.5                   | 1                            |
> | **Complete Graph w/ $d_{ij}$**                      | 59.0 ± 6.6    | 50.0 ± 0.0    | 6.5                   | 1                            |
> | **SCHull w/ $d_{ij}$**                              | **100.0 ± 0.0** | **100.0 ± 0.0** | 2.57                 | 1                            |
> | **SCHull w/ $d_{ij}$ and $\tau_{ij}$**              | **100.0 ± 0.0** | **100.0 ± 0.0** | 2.57                 | 1                            |
>
>
>
> ------
>
> Thank you for considering our rebuttal. We appreciate your feedback and are happy to address further questions on our paper.

---

> > ### Author Response · Authors · 2024-11-22
> > **Thank you**
> >
> > Thank you for considering our rebuttal, and we appreciate your endorsement.

---

### Official Review · Reviewer_SxaB · 2024-11-02

**Soundness:** 3
**Presentation:** 3
**Contribution:** 3
**Rating:** 8
**Confidence:** 3

**Summary:**

The present manuscript proposes a novel graph construction method, SCHull for molecular modeling thorugh GNNs. The authors start by introducing limitations of the currently existing graph construction  methods (e.g. radial cuttoff, and knn graphs), that cannot deal with sparsity, connectivity, and rigidity at the same time.

The method constructs graphs in two steps, first by projecting points onto the unit sphere. and secondly by constructing a convex hull for the projected points, and adding edges to the graph, based on the convex hull. According to the authors' motivation, the resulting graphs are sparse, connected, and rigid, making them suitable for learning molecular representations using GNNs.

The authors show the impact of SCHull, by integrating it into current molecular Gnns, and test it in various benchmarks, including MD17, as well as protein modeling tasks (fold, enzyme reaction classification, Protein-ligand binding).

**Strengths:**

1. I particularly find Theorem 3.6 interesting, and useful for the distinction of point clouds through their corresponding SCHull graphs. In general, the authors provide a well-defined set of propositions for defining connectivity, rigidity, and sparsity for the SChull graphs, making it useful for predicting their impact on GNN's performance and scalability.

2. The motivation for a hyper-parameter free approach for graph construction is very intriguing, and can be proven very useful for molecular modeling tasks, where the distance thresholds can play an important role.

3. The empirical results show that in both small- and large-molecular modeling tasks, SChull has a positive impact in a couple of GNNs.

4. The paper is well-written, the formulations are clear, and the experimentation setup is nicely described.

5. It's very interesting that Tables 4, and 5 show a comparable increase of the runtime cost, with respect to the methods without the SChull. I would expect a much longer runtime discrepancy, due to the point projection, and the CH computation.

5. The code is available, seemingly containing all variants of models and benchmarks appeared in the manuscript. I cannot assess its validity, as I did not run it.

**Weaknesses:**

1.  The paper primarily compares SCHull to radial cutoff and k-nearest neighbor graphs, as well as chemical graphs. It lacks  some compariso with more recent or complex molecular graph construction methods, such as power graphs [1] (which are by the way already mentioned by the authors but not benchmarked) or methods focusing on higher-order graphs, such as simplicial complexes [2] ( that define volumes of data points), or combinatorial complexes [3] (that particularly make use of convex hulls and their corresponding volumes in point clouds).

2. An approach on learning new geometric interactions might be useful in better interpreting the correlation of geometry and function, which could be particularly a motivation for this paper. The paper should focus on exploring or at least mention how such a convex hull approach, despite a comparable computational cost can help on finding interpretable structures.

3. Since the method is focusing on the three properties of sparsity, connectivity, and rigidity, it'd be great if the authors can show empirically the impact of SChull on each of these properties independently.

[1] On the Expressive Power of Sparse Geometric MPNNs. Sverdlov et al 2024.
[2] E(n) Equivariant Message Passing Simplicial Networks. Eijkelboom et al 2023.
[3] E(n) Equivariant Topological Neural Networks. Battiloro et al 2024.

**Questions:**

Based on my comments in the weaknesses section:

1. Can the authors provide a comparison (at least conceptual, but ideally an empirical one) with methods that include power graphs, or higher-order structures?

2. Do the authors think that such a method can provide interpretability benefits? If yes, I think that the motivation could be much strengthened. In that case, is it possible to provide an interoperability analysis?

3. Can the authors show the impact of the method empirically on each graph property mentioned in the paper (I..econnectivity, and rigidity)

---

> ### Author Response · Authors · 2024-11-19
> **Response to Reviewer SxaB (part 1/2)**
>
> We thank the reviewer for the thoughtful review, valuable feedback, and endorsement. In what follows, we provide point-by-point responses to your comments.
>
> ----
>
> **Q1. Can the authors provide a comparison (at least conceptual, but ideally an empirical one) with methods that include power graphs, or higher-order structures? References: [1] On the Expressive Power of Sparse Geometric MPNNs. Sverdlov et al 2024. [2] E(n) Equivariant Message Passing Simplicial Networks. Eijkelboom et al 2023. [3] E(n) Equivariant Topological Neural Networks. Battiloro et al 2024.**
>
> **Response**: We appreciate the reviewer’s suggestion to compare our method with approaches involving power graphs and higher-order structures.
>
> In the revised manuscript, we have included empirical comparisons with power graphs; see Appendix A for its discussion and Tables 6 and 7 in Appendix B for empirical evidence. As highlighted in Remark 3.8, power graphs tend to produce dense graphs and do not resolve issues related to disconnected components. Specifically, the power operation does not alter the number of connected components, and without connectivity, these graphs cannot guarantee rigidity.
>
> Regarding higher-order structures, such as simplicial and combinatorial complexes, we note that they are beyond the primary scope of our paper, which focuses on graphs consisting of only nodes and edges. However, we acknowledge their potential and have included a discussion in Appendix A that discusses their limitations on computational complexity and their possible integration with our graph. Specifically, simplicial complexes, as constructed in [2], rely on radius-based connectivity, which may not guarantee connectivity (and thus rigidity) in all cases. We have demonstrated in Figure 1 that radius graphs struggle to balance sparsity and connectivity, highlighting this limitation. However, it would be interesting to see how our graph construction could serve as input for the method proposed in [2] and how the corresponding simplicial complex would perform.
>
> Combinatorial complexes represent a broader class of higher-order structures, including simplicial complexes. In particular, we find the concept of molecular combinatorial complexes, as introduced in [3], interesting. These complexes incorporate atoms, bonds, rings, and functional groups as cell structures. We believe that finding a set of invariant features that ensure the combinatorial complex exhibits rigidity is a potential direction for future work.
>
> Nevertheless, simplicial complexes and combinatorial complexes often involve high computational complexity in their construction, making them less efficient than our approach, especially on large proteins. We believe it would be worthwhile to investigate whether sparse, connected, and rigid simplicial complexes or combinatorial complexes can be constructed and applied effectively in higher-order models.
>
> We thank the reviewer for bringing these insightful directions to our attention. The revised manuscript includes these discussions and references.
>
> ----
>
> **Q2. An approach on learning new geometric interactions might be useful in better interpreting the correlation of geometry and function, which could be particularly a motivation for this paper. The paper should focus on exploring or at least mention how such a convex hull approach, despite a comparable computational cost can help on finding interpretable structures. Do the authors think that such a method can provide interpretability benefits? If yes, I think that the motivation could be much strengthened. In that case, is it possible to provide an interpretability analysis?**
>
> **Response**: This is an excellent point, and we appreciate the suggestion to explore the interpretability benefits of our method. To strengthen the contribution and highlight its potential impact, we have added a new section in Appendix E that discusses the geometric interpretability of our approach. In particular, we demonstrate the relationship between our SCHull graph and the convex hull, illustrating how the information from our attributed SCHull graph can be used to recover the convex hull of the original point clouds. Additionally, we reference the study [4] that has employed the convex hull to improve predictions of hydrodynamic properties of large molecules, underscoring the practical implications of this interpretability.
>
> [4] Fleming, P. J., & Fleming, K. G. (2018). HullRad: fast calculations of folded and disordered protein and nucleic acid hydrodynamic properties. Biophysical journal, 114(4), 856-869.
>
> ----

---

> ### Author Response · Authors · 2024-11-19
> **Response to Reviewer SxaB (part 2/2)**
>
> **Q3. Can the authors show the impact of the method empirically on each graph property mentioned in the paper (i.e. connectivity, and rigidity)?**
>
> **Response**: Thank you for your suggestion. Combined with the suggestion of Reviewer NBfx, we have now included an empirical comparison of graphs in Appendix B on the counterexamples of symmetric point clouds provided in [5]. In this comparison, we report key graph properties, including the ratio between the number of edges and the number of nodes (sparsity) and the number of connected components, across various graph construction methods, such as radius graphs, power graphs, complete graphs, and our SCHull graph. We have also tested the performance of GNNs on these graphs and reported the mean accuracy and standard deviation over 10 independent runs. While performance does not directly reflect rigidity, it serves as a good indicator of the potential for rigidity.
>
> From the experiments, we observe that our SCHull method outperforms the other methods in terms of sparsity, connectivity, and without the need for hyperparameter tuning. Additionally, it enables GNNs to distinguish between non-isomorphic symmetric point clouds, which we interpret as an indication of greater rigidity. These results highlight the effectiveness of SCHull in capturing essential geometric properties, even for non-generic point clouds.
>
> [5] Li, Zian, et al. Is distance matrix enough for geometric deep learning? NeurIPS 2024.
>
> ------
>
> Thank you for considering our rebuttal. We appreciate your feedback and are happy to address further questions on our paper.

---

> > ### Comment · Reviewer_SxaB · 2024-11-22
> >
> > I'd like to thank the reviewers for their detailed response!
> > All of my comments have been addressed successfully.

---

> > > ### Author Response · Authors · 2024-11-22
> > > **Thank you**
> > >
> > > Thank you for your active feedback and engagement during the rebuttal process. We appreciate your endorsement!

---

### Comment · Area_Chair_jAiS · 2024-11-22

Dear reviewers,

The authors have submitted their rebuttals. If you haven't already, please kindly review their responses and participate in the discussions. Additionally, please indicate if and how their responses have changed your opinions.

Thank you,

AC

---

### Public Comment · ~Kun_Li10 · 2025-05-02
**Github is MISSING**

Hi, your work is great, I want to run it! Where is your true GitHub URL?

---

> ### Public Comment · ~Shih-Hsin_Wang1 · 2025-05-02
> **Github URL**
>
> Thank you for your support! We’ve updated our GitHub repository URL. Please visit: https://github.com/shihhsinwang0214/SCHull

---

> > ### Public Comment · ~Kun_Li10 · 2025-05-03
> > **Thank you for informing.**
> >
> > Thank you for informing.

---

> > ### Public Comment · ~Kun_Li10 · 2025-05-05
> >
> > There seem to be some missing files or potential logical issues in the current version of your code ( https://github.com/shihhsinwang0214/SCHull), which make it difficult to run successfully on my side. It would be greatly appreciated if you could share a tested and fully functional version when possible.

---

> > > ### Public Comment · ~Shih-Hsin_Wang1 · 2025-05-29
> > > **Response**
> > >
> > > Our server experienced a crash, but we’ve just recovered it. We're currently working on cleaning up the experimental code before making it publicly available. We apologize for the inconvenience and expect to release everything as soon as possible—hopefully within a week. We appreciate your patience!

---

### Meta-Review · Area_Chair_jAiS · 2024-12-20

**Metareview:**

The paper studies 3D graph construction, which is an important yet somehow overlooked problem in the field of 3D GNNs. The proposed method SCHull first projects all points onto the unit sphere and then constructs a convex hull for the projected points. Overall, the work is strongly motivated and technically novel, and all reviewers hold positive opinions towards the paper. Thus, an acceptance is recommended. I also read through the paper, and suggest the authors rigorously define Rigidity in the final version of the paper. Additionally, based on my understanding, the final set of edges is a union of the SCHull results and cut-off results (similar to existing works where the cutoff is usually large like 6A). If this is true, I hardly see how the sparsity is fullfilled especially for empirical studies.

**Additional Comments On Reviewer Discussion:**

There were effective communications between the authors and reviewers, and seems almost all the concerns have been successfully addressed.

---

### Decision · Program_Chairs · 2025-01-22

Accept (Oral)